# Interactive impacts of meteorological and hydrological conditions on the physical and biogeochemical structure of a coastal system

Onur Kerimoglu[1,2], Yoana G. Voynova[1], Fatemeh Chegini[3,4], Holger Brix[1], Ulrich Callies[1], Richard Hofmeister[1], Knut Klingbeil[3], Corinna Schrum[1], and Justus E.E. van Beusekom[1]

[1]Institute for Coastal Research, Helmholtz-Zentrum Geesthacht, Germany
[2]Institute for Chemistry and Biology of the Marine Environment, Carl von Ossietzky University Oldenburg, Germany
[3]Leibniz Institute for Baltic Sea Research, Warnemünde, Germany
[4]now at: Max Planck Institute for Meteorology, Hamburg, Germany

**Correspondence:** Onur Kerimoglu (kerimoglu.o@gmail.com)

**Abstract.** The German Bight was exposed to record high riverine discharges in June 2013, as a result of flooding of the Elbe and Weser rivers. Several anomalous observations suggested that the hydrodynamical and biogeochemical states of the system were impacted by this event. In this study, we developed a biogeochemical model and coupled it with a previously introduced high resolution hydrodynamical model of the southern North Sea, in order to better characterize these impacts, and gain insight into the underlying processes. Performance of the model was assessed using an extensive set of *in-situ* measurements for the period 2011-2014. We first improved the realism of the hydrodynamic model with regard to the representation of cross-shore gradients, mainly through inclusion of flow-dependent horizontal mixing. Among other characteristic features of the system, the coupled model system can reproduce the low salinities, high nutrient concentrations and low oxygen concentrations in the bottom layers observed within the German Bight following the flood event. Through a scenario analysis, we examined the sensitivity of the patterns observed during July 2013 to the hydrological and meteorological forcing in isolation. Within the region of freshwater influence (ROFI) of the Elbe-Weser rivers, the flood event clearly dominated the changes in salinity and nutrient concentrations, as expected. However, our findings point out to the relevance of the peculiarities in the meteorological conditions in 2013 as well: a combination of low wind speeds, warm air temperatures and cold bottom water temperatures resulted in a strong thermal stratification in the outer regions, and limited vertical nutrient transport to the surface layers. Within the central region, the thermal and haline dynamics interactively resulted in an intense density stratification. This intense stratification, in turn, led to enhanced primary production within the central region enriched by nutrients due to the flood, but reduction within the nutrient-limited outer region, and it caused a wide-spread oxygen depletion in bottom waters. Our results further point to the enhancement of the current velocities at the surface as a result of haline stratification, and intensification of the thermohaline estuarine-like circulation at the Wadden Sea, both driven by the flood event.

# 1 Introduction

Riverine discharges influence the thermohaline stratification, nutrient availability and as a result, primary production within the coastal zones (e.g., Hickey et al., 2010; Cloern et al., 2014; Emeis et al., 2015). Excess amounts of riverine nutrient inputs cause coastal eutrophication, associated with a host of problems (Smith and Schindler, 2009), including development of dense and harmful algal blooms (e.g., Garnier et al., 2019), decline of submerged vegetation (e.g., Dolch et al., 2013), and oxygen depletion (see the review by Fennel and Testa, 2019). Fraction of riverine freshwater and nutrients that reaches the open ocean is an open question, with estimates ranging between 15% and 80 % (Sharples et al., 2017; Izett and Fennel, 2018).

Mixing of riverine freshwater with the surrounding saline marine waters at the coasts is driven by a set of hydrodynamical processes intriguingly linked together (for a recent review, see Geyer and Maccready, 2014). The freshwater inputs by rivers may lead to haline stratification in the coastal region, in the absence of any thermal stratification (van Aken, 1986). Horizontal density gradients caused by riverine freshwater inputs govern gravitational circulation (i.e., exchange flow), where the seaward flow of the lighter water at the surface is counteracted by a landward flow of the saltier, denser waters near the sea floor (see Burchard et al., 2018). Destabilizing and stabilizing effects of flood and ebb currents, respectively, may further enhance the gravitational circulation (Burchard and Hetland, 2010).

The study system, the German Bight, is a shallow area located in the southeastern North Sea (Fig. 1). The prevailing wind direction is southwesterly (Siegismund and Schrum, 2001), governing a large cyclonic gyre within the southern North Sea (Sündermann and Pohlmann, 2011). But under easterly and northeasterly winds, anticyclonic circulation may prevail (Becker et al., 1992; Dippner, 1993; Callies et al., 2017). Occurrence of thermohaline stratification within the German Bight is driven by buoyancy inputs from the rivers to the coastal waters and heat fluxes in deeper areas (Frey, 1990; Simpson et al., 1993). Stratification is also strongly influenced by wind intensity and direction: while westerly winds allow, and easterly winds enhance stratification, southerly winds have a particularly destratifying effect (Schrum, 1997). An estuarine-like circulation has been shown to be present within the coastal areas of the German Bight (Burchard et al., 2008; Burchard and Badewien, 2015). This mechanism has been suggested to contribute to the maintenance of the steep, cross-shore suspended particulate matter (SPM) and nutrient gradients (Flöser et al., 2011; Hofmeister et al., 2017), with regional differences (van Beusekom et al., 2019). The steep cross-shore gradients observed in SPM and nutrient concentrations have been recently reproduced by numerical models (Staneva et al., 2009; Gräwe et al., 2016) owed to high resolution grids and the terrain-following vertical coordinates that enable representation of the estuarine circulation.

Surrounded by industrialized and densely populated countries, the southern North Sea has been experiencing eutrophication related problems (Radach, 1992; Hickel et al., 1993; OSPAR, 2017), such as occasional oxygen depletion events during summer (Frey, 1990; Große et al., 2016). The Elbe and Weser rivers have been estimated to be the primary sources of nitrogen (N) in the southern North Sea (Große et al., 2017). Since the 1980s, nutrient concentrations in these and other contributing rivers (e.g., Rhine, Meuse), have been significantly reduced, more for phosphorus (P) than for N (Radach and Pätsch, 2007). This has resulted in some improvement in water quality, especially within the northern Wadden Sea (Wiltshire et al., 2008; van

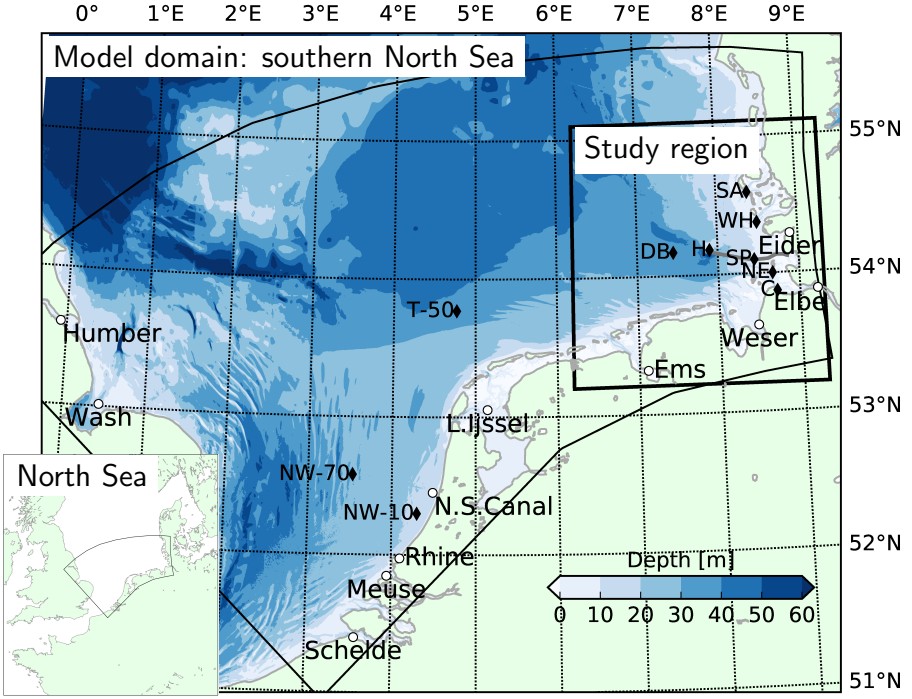

**Figure 1.** Model domain, bathymetry (data from the European Marine Observation and Data Network, EMODnet), and the location of the study region, the German Bight. Filled circles: location of river mouths on the grid, diamonds: monitoring stations (NW: Noordwijk, T: Terschelling, DB: Deutsche Bucht, H: Helgoland, SP: Suederpiep, SA: Southern Amrum, WH: Westerhever, NE: Norderelbe, C:Cuxhaven), gray line: the average route of the Ferrybox transect between Helgoland and Büsum (see section 2.3).

Beusekom et al., 2019), but according to a recent study, the nutrient concentrations within the coastal areas are estimated to be
still 50-70% higher than the pre-industrial levels (Kerimoglu et al., 2018).

The extent to which the hydrodynamical structure, and the transport of riverine material within the German Bight depends on the interannual variability in riverine discharges is not fully understood. In particular, whether and to what extent a flood event would influence the thermohaline stratification within the off-shore waters, or the estuarine circulation at the coastal waters has not been explicitly investigated. In this study, based on the simulations obtained with a coupled physical-biogeochemical model,
we examine the physical and biogeochemical structure in the German Bight during July 2013, i.e., following a major flood event (Voynova et al., 2017), in comparison to those in the previous year, to characterize the sensitivity of the hydrodynamical and biogeochemical structure within the German Bight to the meteorological and hydrological conditions. Through a numerical scenario analysis, we try to disentangle the effects of the flood event, meteorology, and in particular the wind conditions.

## 2 Material & Methods

### 2.1 The Model

The hydrodynamical host, the General Estuarine Transport Model (GETM, Burchard and Bolding, 2002) is a free-surface baroclinic model that uses terrain-following vertical coordinates. GETM was previously applied to the greater North Sea area (Stips et al., 2004; Pätsch et al., 2017) and the German Bight and Wadden Sea regions in higher resolution (Staneva et al., 2009; Gräwe et al., 2016). In the current application, GETM is defined on a curvilinear grid with a resolution of 1.5-4 km (Fig. 1) and 20 vertical layers, and operated with integration time steps of 5 s and 360 s for the barotropic and baroclinic modes, respectively. At the northern and western boundaries, surface elevations extracted from TRIM-NP-2D (Gaslikova and Weisse, 2013) are provided as clamp boundary conditions at hourly resolution (see below for other boundary conditions). For the discretization of advection, we employed a third order, total variation diminishing P2-PDM (i.e., ULTIMATE QUICKEST) scheme, recognized for its accuracy and gradient conserving qualities (e.g., Pietrzak, 1998; Burchard and Rennau, 2008).

An almost identical model setup was previously employed and shown to capture the spatial and temporal distributions of temperature and salinity within the German Bight for the period 2000-2010 (Kerimoglu et al., 2017a), as well as the tidal dynamics (Nasermoaddeli et al., 2018). Since then, the following refinements were made: *i*) providing meteorological forcing at hourly resolution extracted from a COSMO-CLM hindcast simulation (Geyer, 2014), which was previously at 6-hr resolution; *ii*) specifying the monthly average vertical temperature and salinity profiles at the boundaries for each year separately as predicted by HAMSOM (for a recent description of the setup, see Große et al., 2017), instead of providing climatological averages for all years; *iii*) explicitly describing horizontal diffusion through a Smagorinsky parameterization (Smagorinsky, 1963). Impacts of the first two refinements *i-ii* were subtle and local, but introduction of horizontal diffusion (*iii*) systematically improved the representation of coastal gradients, and resulted in more plausible total mixing rates overall (see Appendix A).

The biogeochemical model employed here, provisionally named 'Generalized Plankton Model' (GPM), has been recently developed. It has two main components: a component that describes plankton dynamics, and a geochemistry component that describes the recycling of the organic material within the water and sediments. These compartments, both of which are implemented as FABM (Framework for Aquatic Biogeochemical Models, Bruggeman and Bolding (2014)) modules, are coupled in run-time. Elemental fluxes between various model compartments are illustrated in Fig. 2.

The plankton component has been developed based on the carbon (C-) and P- resolving generic plankton model, described by Kerimoglu et al. (2017b) in the context of a lake application. Specifically, the extensions included descriptions of N and silicate (Si) limitation of phytoplankton (diatoms for the latter), and variations of the Chl:C ratio according to Geider et al. (1997). Heterotrophs can now handle and properly recycle prey with constant or variable C:N:P:Si ratios. The 'genericity' of the previous model version (Kerimoglu et al., 2017b) was due to the fact that each plankton species was described as a potential mixotroph with a prescribed autotrophy/heterotrophy ratio. In the new version, explicit phytoplankton and zooplankton modules are used, in order to facilitate future development, where phytoplankton-, zooplankton- and mixotroph- specific functionalities are foreseen to be included in future work. In the present application, plankton comprises of two phytoplankton functional groups, namely diatoms and flagellates, and two zooplankton functional groups, namely micro- and meso-zooplankton.

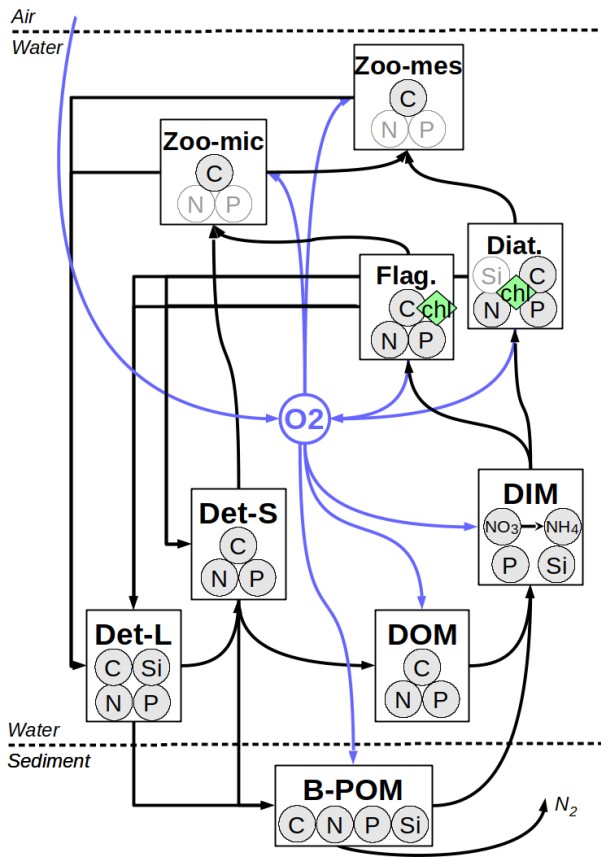

**Figure 2.** Elemental fluxes between model compartments. Det-L and Det-S: large and small detritus, DOM: Dissolved Organic Matter, DIM: Dissolved Inorganic Matter, B-POM: Benthic Particulate Organic Matter. The pale N and P in micro- and mesozooplankton and Si in diatoms represent diagnostic state variables which are determined by a fixed prescribed ratio to the C-bound to these pools, resolved as a state variable. For the sake of simplification, fluxes from phytoplankton and zooplankton to DOM and DIM pools are not shown (see Appendix B for a detailed description of model).

The abiotic component (i.e., module describing non-planktonic processes) is largely based on ECOHAM. Description of the dynamics of two detritus pools (large and small), dissolved oxygen, dissolved inorganic material (DIM) pool that resolves $PO_4$,
Si, $NO_3$ and $NH_4$, and a plate (not vertically resolved) benthic pool are as in Lorkowski et al. (2012). ECOHAM's carbonate cycle was excluded, and a simpler description for DOM remineralization was used. Finally, light conditions are determined by the shading by phytoplankton, detritus, DOM and a parameterization of background turbidity caused by SPM. A detailed description of the model formulations and parameters can be found in Appendix B.

Starting from the initial conditions obtained earlier, the model was spun-up for the period 2008-2010 with the parameteriza-
tion presented here, since up to 3 years was found to be necessary for the solutions to converge from arbitrary initial conditions. We then considered the period 2011-2014 for the model performance assessment. For the analysis of the years 2012 and 2013,

in addition to the reference run, we consider three scenarios in order to investigate the sensitivity of the physical and biogeo-chemical structure of the system to the meteorological and hydrological forcing: based on the 2013 run (with respect to ocean boundary and initial conditions), the scenario '2013-R12' was run with the river forcing of 2012, '2013-M12' was run with the meteorological forcing of 2012. In a third scenario, '2013-W12', only June-August 2013 was simulated with the wind and atmospheric pressure fields from the respective months in 2012, starting from the initial conditions of June 2013 and using the ocean boundary conditions of 2013.

## 2.2 Riverine and Atmospheric Forcing

Both for atmospheric forcing of the coupled physical-biogeochemical model, and for the analysis of meteorological condi-tions, we use the COSMO-CLM atmospheric hindcast, that has a 0.22° resolution (Geyer, 2014). Meteorological forcing from COSMO-CLM comprises precipitation, total cloud cover, mean sea level pressure, relative humidity and air temperature at 2m above sea surface, and U- and V- components of wind at 10m above sea surface, whereas evaporation was calculated by GETM. Shortwave radiation at the surface was calculated according to astronomical functions provided by GETM, and corrected by cloud cover and seasonal variations in surface albedo according to Payne (1972). Longwave radiation was calculated according to Clark et al. (1974). Momentum and heat fluxes were calculated according to bulk formulae by Kondo (1975).

Atmospheric deposition rate of oxidized and reduced nitrogen, added respectively to the modelled $NO_3$ and $NH_4$ at the surface layer, was downloaded from the website of the European Monitoring and Evaluation Programme (EMEP). Riverine discharges and nutrient fluxes were derived from the OSPAR Comission's ICG-EMO (Intersessional Correspondence Group on Eutrophication Modelling) database, provided by S. van Leeuwen (NIOZ) upon personal request. Here, we considered only the major rivers shown in Fig. 1 (Witham, Welland, Nene and Great Ouse are collectively labeled as the 'Wash'). Based on Amann et al. (2012), 30 % of the organic material (total minus inorganic form for each of the C, N, P and Si) is assumed to be in particulate form (detritus), and the rest to be in dissolved form (DOM). Small (<30d) gaps in riverine data were filled using linear interpolation, and larger gaps were replaced with long-term (2000-2017) climatologies. Riverine inputs were applied over the full depth, given the fact that the outlets of all considered rivers are at shallow sites (Fig. 1).

## 2.3 Observation data

Station data (Helgoland, Cuxhaven, Deutsche Bucht; see Fig. 1) for temperature, salinity and oxygen (the latter only at Deutsche Bucht) were downloaded from the COSYNA (Coastal Observing System for Northern and Arctic Seas) data portal (www.cosyna.de, see Breitbach et al., 2016) at daily resolution (snapshots at 00:00 averaged within an hourly time window). Collection and processing of the semi-continuous data collected by FerryBox platforms at the Cuxhaven and Helgoland mon-itoring stations and on the M/V *Funny Girl* ferry operating between Büsum and Helgoland during May-September have been described previously by Petersen et al. (2011) and Voynova et al. (2017) and are available from the COSYNA data portal as well.

N, P, Si and chlorophyll data at the Helgoland-Roads station were collected at semi-daily (every working day), and using standard procedures as described by Wiltshire et al. (2008). Data from the Noordwijk, Terschelling, Norderelbe, Suederpiep

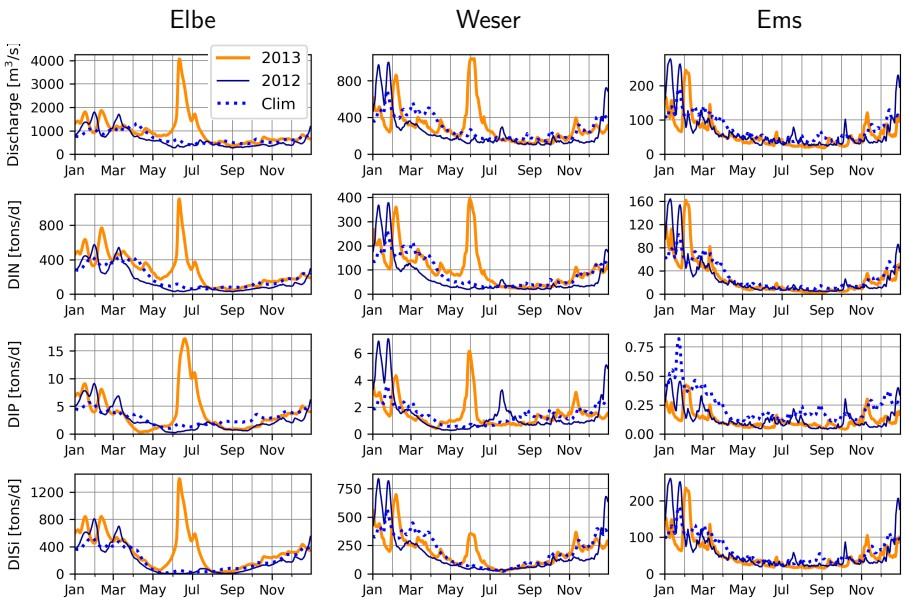

**Figure 3.** Measured discharge, DIN (NO$_3$+NH$_4$), DIP and DISi loading rates at rivers Elbe, Weser, and Ems during 2012 (dark blue lines), 2013 (orange lines) and the 2005-2014 climatology, excluding 2013 (dashed blue lines).

and Westerhever stations are available at monthly intervals. For the Noordwijk-70 and Terschelling-50 stations, we consider only the surface measurements available at biweekly intervals, while the data at other stations are located at shallow sites, and therefore provide only surface measurements. Mooring data for surface (<10 m) salinity, temperature and nutrients, randomly distributed over the entire model domain and simulation period 2011-2014, were obtained from the International Council for the Exploration of the Sea (ICES). In this dataset, the outliers, defined as the values falling outside the $[\bar{o} \pm 4\sigma]$ range, where $\bar{o}$ and $\sigma$ stand for the mean and standard deviation of the raw observations, were removed.

Spatial matching of all data was performed by calculating the distance-weighted mean of the nearest four modelled grid values around the observation, using the 'spatial.cKDTree' package from the Scipy library, version 1.1.0 of Python 3.5.

## 3 Results

### 3.1 Hydrological and Meteorological Conditions

Variability of discharge rates and concentration of inorganic and organic constituents for the period 1977-2000 was explored by Radach and Pätsch (2007). Here, we analyze the discharge and nutrient fluxes for the specific time period of interest. Typically, the discharge rates of the continental rivers around the southern North Sea peak during winter/early spring (e.g., Lenhart et al., 1997). For the rivers Elbe, Weser and Ems, the major rivers discharging to the German Bight, this pattern holds for the decade that includes and precedes the time period of interest, and 2012 in particular (Fig.3). But during June 2013, a large

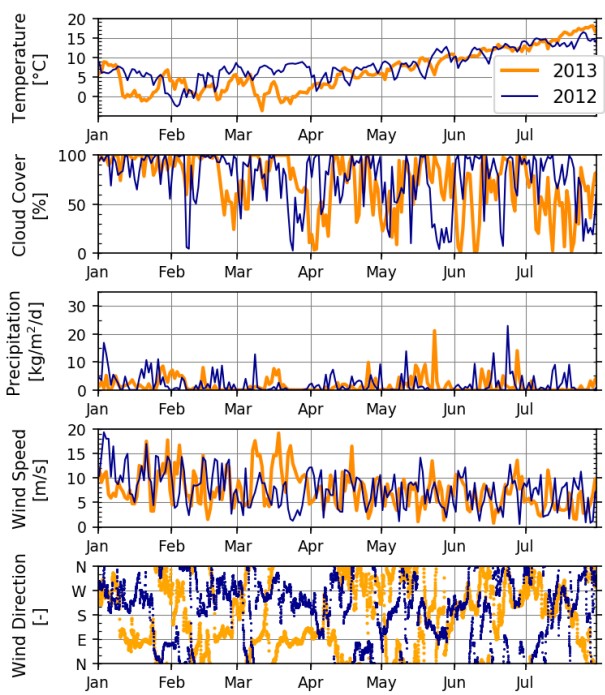

**Figure 4.** Meteorological variables during January-July 2012 and 2013, extracted from a representative grid point (54°14'N, 7°29'E) of the meteorological hindcast, used also as model forcing (see Section 2.2). Temperature is from 2 m and wind is from 10 m height above the sea level. Wind direction is shown at hourly resolution, all other variables at daily resolution.

precipitation event over central Europe caused flooding of several major river basins (Merz et al., 2014), including Elbe and Weser (Fig.3). The Elbe flood can be considered as a 100-year event with discharge rates of up to 4060 m$^3$ s$^{-1}$ during 11 and 12 June (Voynova et al., 2017), which is four-fold higher than the typical discharge rates during winter (Fig.3). Ems, and the other rivers in the model domain do not show such an extreme response, underlining the locality of the aforementioned meteorological event. Nitrogen, phosphorus and silicate concentrations did not vary systematically during the flood event, and therefore their fluxes paralleled the discharge rates, with distinct peaks during June 2013 for the Elbe and Weser rivers (Fig.3).

Meteorological conditions during 2012 and 2013 differed systematically during two periods (Fig. 4). The first of these occurred during the early spring: March 2012 was characterized by relatively warm air temperatures and winds mildly blowing from the west/southwest, whereas March 2013 was cold with strong easterly winds. The second period occurred during the middle of summer: July 2012 was relatively cold with overcast skies and some precipitation, contrasting with warmer, drier and calmer conditions in July 2013.

## 3.2 Assessment of the Model Performance

Simulated temperature and salinities at the surface match, in general, well the observations found in the ICES database, randomly distributed throughout the model domain within the period 2011-2014 (Fig. 5). There is a slight cold bias at the lower temperature range (29-32 g/kg), which seems to be canceled out by the slight warm bias at the higher range. These deviations

are mostly within a 1 °C range, therefore presumably do not have a significant effect. At an intermediate range, salinity is overestimated by up to 2 g/kg, indicating insufficient spread of coastal waters with low salinity. This may either be due to (still) underestimated horizontal mixing (see Appendix A), or inaccuracies in the advection patterns. Match of the simulated $NO_3$ and DIP to the ICES-observation set is reasonably well, with -5% normalized bias and correlation coefficients larger than 0.6 for both variables (Fig. 5). Underestimated $NO_3$ at an intermediate range (10-40 $\mu$MN) is possibly due to the aforementioned

underrepresentation of N-rich riverine waters within the transition zone. Regarding DIP, the measured-simulated pairs that represent major underestimation errors (e.g., in the <0.5 $\mu$MP simulation band) point to the inability of the model to capture summer maxima occurring in specific coastal regions.

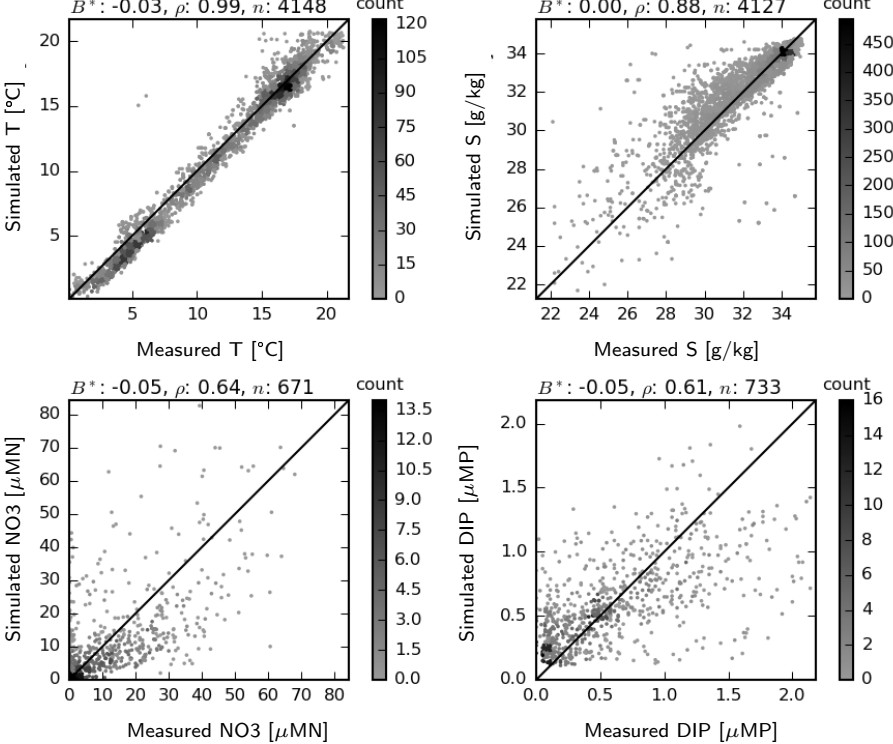

**Figure 5.** Two dimensional histogram of simulated vs measured temperature, salinity, $NO_3$ and DIP at the surface for the period 2011-2014. Count indicates the occurrence frequency of simulation-observation pairs. $B*$: normalized bias, $\rho$: correlation coefficient, $n$: number of observation-simulation pairs.

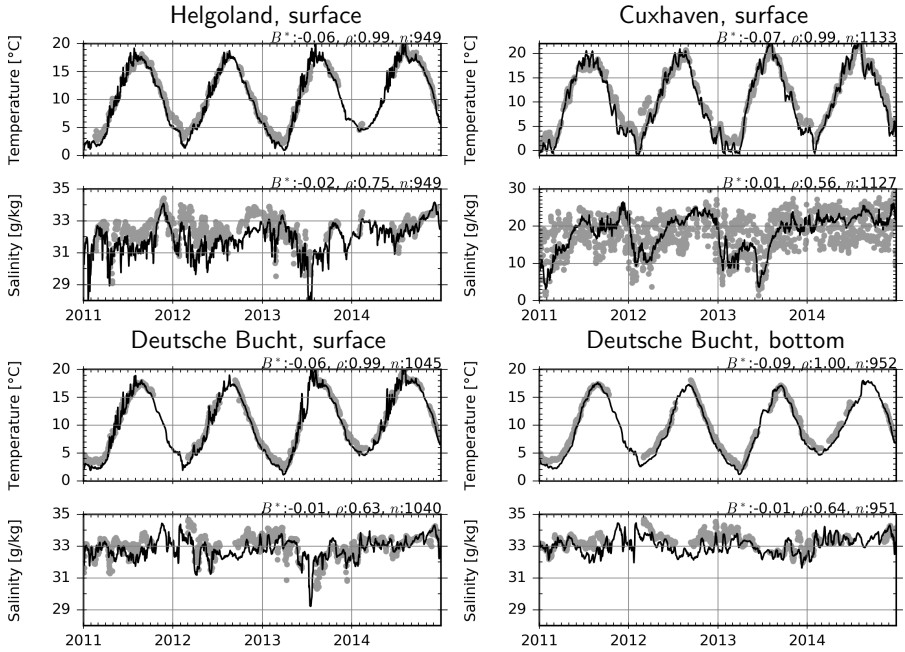

**Figure 6.** Observations (dots) and model estimates (lines) of temperature and salinity. $B*$: normalized bias, $\rho$: correlation coefficient, $n$: number of observation-simulation pairs.

Simulated and measured temperature and salinity are compared at 3 fixed monitoring stations (Fig. 6). Two of these stations, Helgoland and Cuxhaven are located at shallow sites, therefore provide only surface measurements, whereas the third one, the Deutsche Bucht, provides measurements also at 30m depth. At all these stations, temperature is estimated with 5-9% negative bias, and correlation scores range between 0.99-1.0. The interannual variations are well captured: the relatively warm winters (January-March) of 2012 and 2014, and the cold winter of 2013 manifest as cold and warm water temperatures according to the observations, and these differences are realistically reproduced by the model, despite the modeled temperatures being about 0.5-1.0 °C lower. Salinity is modelled consistently with only up to 2% bias at all 3 stations, despite the lower correlation coefficients in comparison to temperature (Fig.6). The relatively higher variability of the salinity measurements is due to the tidal variations (most obvious at the Cuxhaven station), which are smeared out in the daily average model output. The freshwater plume of the flood event of June 2013, and other similar events have been accurately reproduced by the model.

According to the June-July average salinities measured by FerryBox on the M/V *Funny Girl* ferry between Büsum and Helgoland (Fig. 1), the salinities gradually decrease from about 32 g/kg at Helgoland to about 27 g/kg at Büsum in 2012 (Fig. 7). In 2013, driven by the freshwater plume of the flood, the average salinities were lower at both edges, with about 27-29 g/kg at Helgoland and 22-23 g/kg at Büsum. The model estimates are quite accurate at the off-shore areas, but underestimate the observations near the coast, up to 2 g/kg in 2012 and 3-5 g/kg in 2013. Despite these biases, the clear difference between the two years as captured by the FerryBox is qualitatively captured by the model.

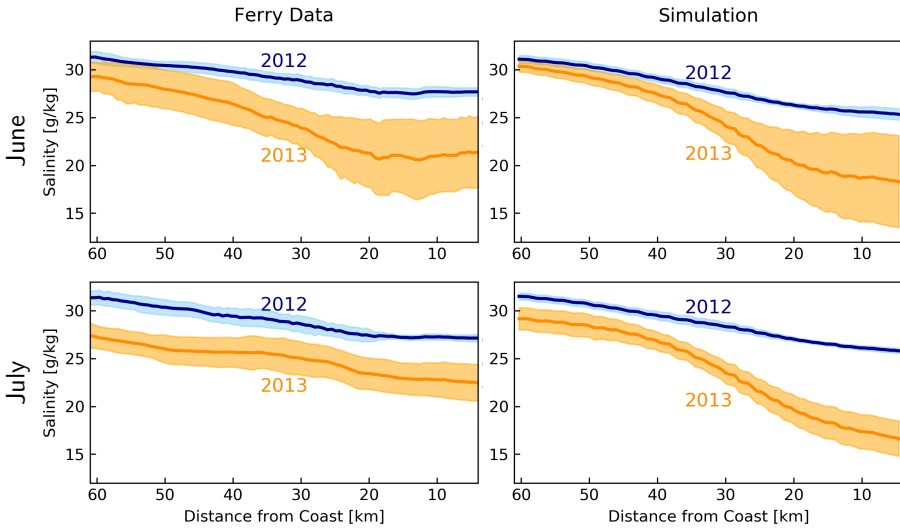

**Figure 7.** Average (dark lines) and standard deviation (shadings) of salinities between Helgoland and Büsum, according to the FerryBox data from M/V *Funny Girl* and simulation by the model during June and July 2012 and 2013.

Dissolved inorganic N (DIN, which in our model comprise $NO_3$ and $NH_4$, as $NO_2$ was not considered) and P (DIP, i.e., $PO4$) are generally well reproduced at all considered monitoring stations (Fig.8), as suggested by low bias and moderate correlations. For dissolved silicate, Si, model estimates overshoot the observations by about 50% at Helgoland and up to 100% at the Noordwijk stations. The latter is mainly driven by the strong DISi fluxes from the western boundary, reflecting the overestimation of Si specified at the boundaries (Fig. 1).

For chlorophyll, there is up to 120% positive bias at the off-shore stations (Fig.8), while the correlation coefficients are particularly low at the Terschelling-50 and Noordwijk-70 stations and moderate at Helgoland and Noordwijk-10. A consistent source of error seems to be the failure of the model to estimate the timing of the spring bloom. However, differences between stations, i.e., values at Helgoland and Noordwijk-10 being higher than at Terschelling-50 and Noordwijk-70 stations, are well reproduced.

Measured and simulated $NO_3$ and DIP concentrations at 3 coastal stations, Norderelbe, Suederpiep and Westerhever, located along the North-Frisian Wadden Sea (Fig. 1), are shown in Fig. 9. For $NO_3$, measurements in both June and July 2013 were distinctly higher than those in 2012 at Norderelbe and Suederpiep stations, but not at Westerhever in July. Despite a tendency to overestimate, the range of simulated values mostly encloses the measurements, and the qualitative differences between 2012 and 2013 and between different stations were captured by the model. Average DIP measurements did not differ between 2012 and 2013, but gradually decreased with distance from the Elbe mouth. The model captures this gradual decline, but the difference it suggests between the two years at the Norderelbe and Suederpiep stations in July is larger than that indicated by the measurements.

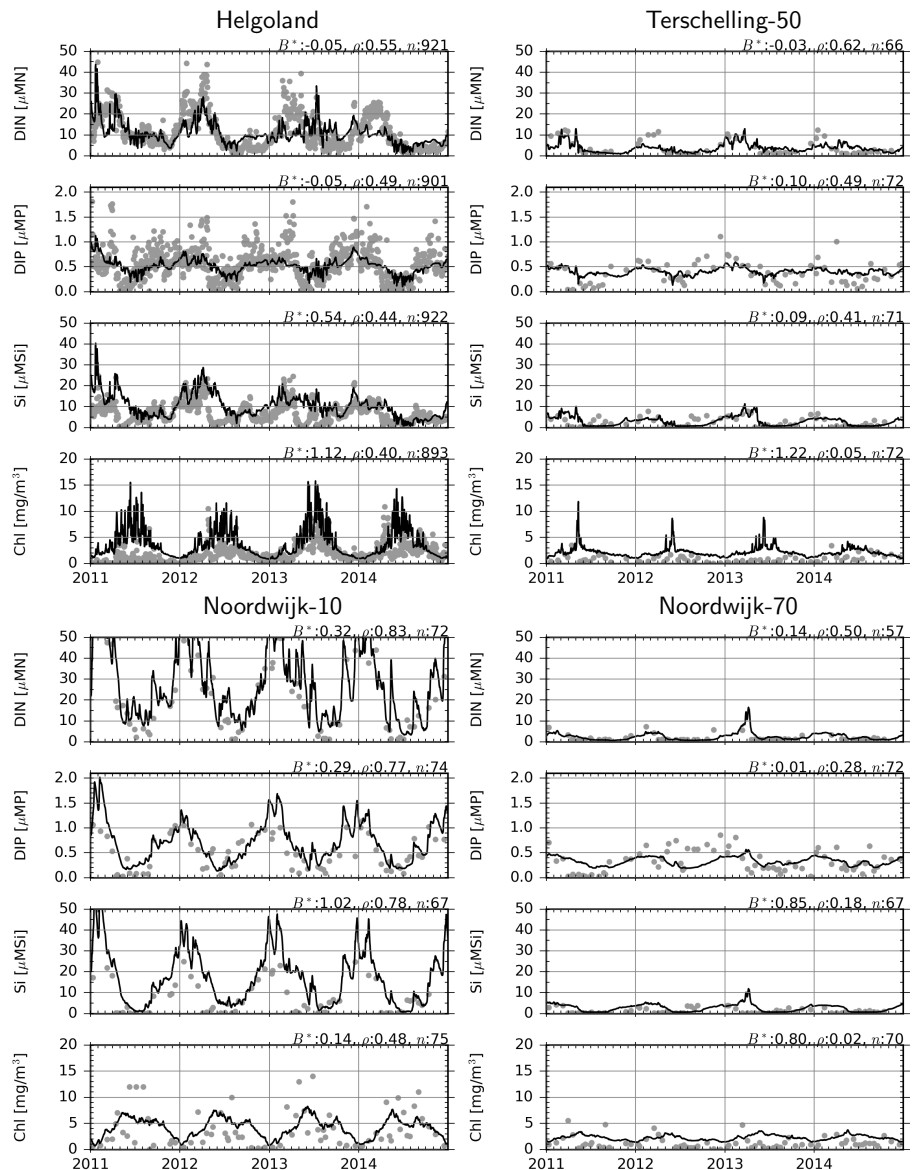

**Figure 8.** Observations (dots) and model estimates (lines) of surface DIN, DIP, DISi and chlorophyll concentrations. $B*$: normalized bias, $\rho$: correlation coefficient, $n$: number of observation-simulation pairs.

## 3.3 Thermohaline structure, nutrient status and productivity of the system

Average salinities in the surface and bottom layers estimated by the model suggest considerable extension of the Elbe-Weser ROFI during July 2013, in comparison to July 2012 (Fig. 10). This extension is similar in surface and bottom layers within the well mixed shallow areas, but stronger at the surface in deeper regions where a thermohaline stratification develops (Fig. 11).

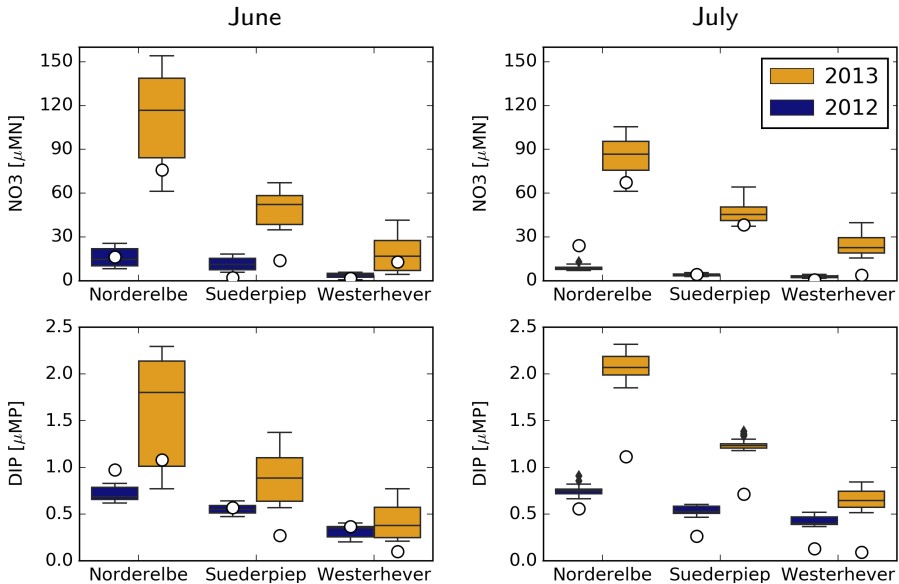

**Figure 9.** Monthly average measurements (circles) and temporal distribution of the simulations (boxes showing the median, 1st and 3rd quartile and whiskers showing the minimum and maximum values) for surface $NO_3$ and DIP concentrations at three coastal stations shown in Fig. 1.

The surface and bottom temperatures display similar horizontal gradients during July 2012 and 2013 with higher temperatures near the coast, and lower temperatures within the offshore regions (Fig. 10). However, the surface temperatures within the outer areas during July 2013 are 1-2 °C higher than those during July 2012 (Fig. 4). In contrast, the bottom temperatures during 2013 July are lower than those during July 2012.

When the riverine forcing of 2012 was used for simulating 2013 ('2013-R12' scenario), the characteristic freshwater plume of 2013 disappears (Fig. 10). The resulting freshwater front (e.g., as hinted by 30 g/kg isohaline) differs from that of 2012 as well, having retreated to the southern latitudes. Under this scenario, the temperatures at the bottom layers remain identical to those of 2013, but the surface layer becomes slightly colder. The latter is explained by the increasing stability of the water column due to the extra buoyancy caused by the flood event in 2013, reflected by the larger area of intense ($>1$ kg m$^{-3}$) density
stratification (Fig. 11, compare 2013 and 2013-R12).

The effect of exchanging the entire meteorological forcing (as indicated by the 2013-M12 scenario), compared to that of exchanging only the short-term (i.e., starting from June) wind forcing (2013-W12 scenario) on the salinity distribution is almost identical: according to both scenarios, the freshwater plume around the mouth of Elbe and Weser is preserved, but the plume spreads along the coast instead of spreading towards the outer German Bight as was the case in the original 2013 simulation
(Fig. 10). Thus, it can be concluded that the distribution of salinity within the central and outer German Bight in July 2013 is driven by the short-term wind conditions. The freshwater front (e.g., as indicated by the 27-30 g/kg isohalines), simulated

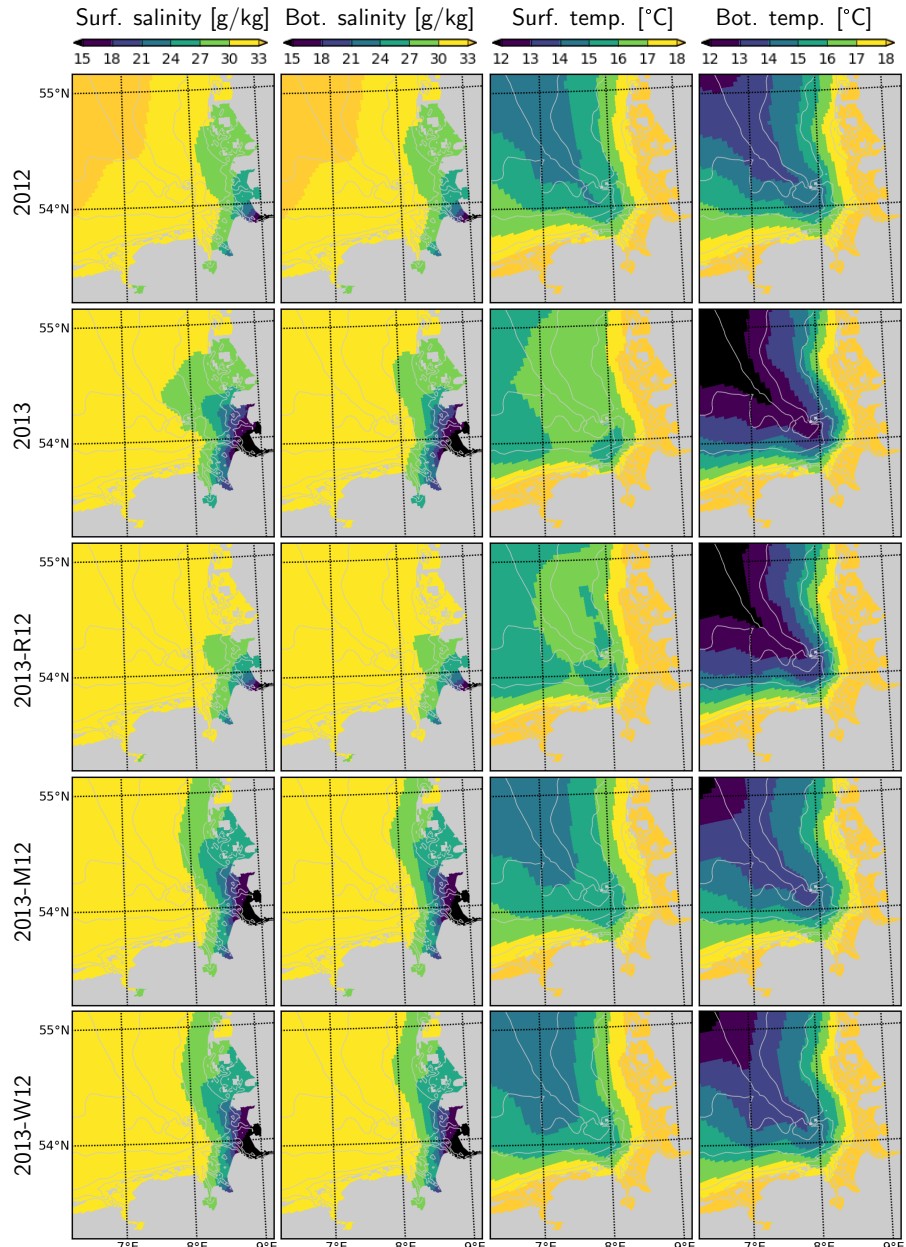

**Figure 10.** Salinity and temperature in the surface and bottom layers during July for the years 2012 and 2013 and scenarios 2013-R12, 2013-M12 and 2013-W12.

according to both 2013-M12 and 2013-W12 scenarios, extends further to the North in comparison to 2012, which is evidently driven by the additional freshwater inputs due to the flood.

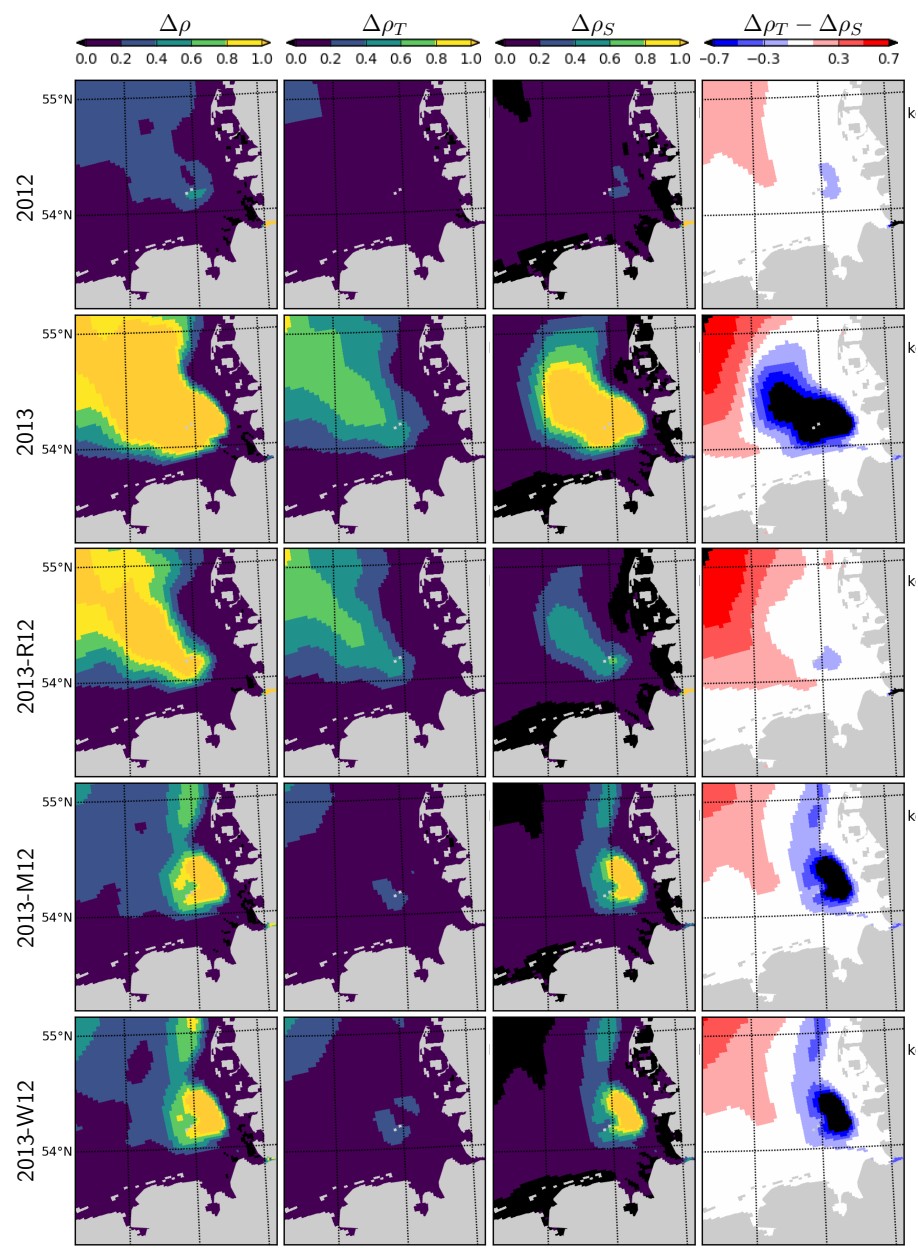

**Figure 11.** Density difference between the surface and bottom layers ($\Delta\rho$), contribution of temperature and salinity, ($\Delta\rho_T$ and $\Delta\rho_S$, see the text), and their difference ($\Delta\rho_T$ - $\Delta\rho_S$), during July for the years 2012 and 2013 and scenarios 2013-R12, 2013-M12 and 2013-W12.

Temperatures simulated according to 2013-M12 scenario are similar to those simulated for 2012, characterized by relatively low temperatures at the surface and the relatively high temperatures at the bottom, in comparison to the original estimations for 2013. Interestingly, the temperatures simulated by the 2013-W12 scenario are similar to those simulated by the 2013-M12

scenario, indicating that the large differences in surface and bottom temperatures during July 2013 were mainly caused by the wind conditions. In the 2013-W12 scenario, enhanced turbulent vertical mixing, driven by the stronger winds in July 2012 does not allow the surface temperatures to build up, while it causes the cold bottom temperatures to increase to the levels originally simulated for July 2012, except within the northwestern margin of the study region, where the bottom temperatures remain cold.

The combination of temperature and salinity dynamics determines the 3-dimensional density ($\rho$) structure of the system. The difference between the density of the surface and bottom layers ($\Delta\rho$) therefore indicates the intensity of the thermohaline stratification, and hence, gives insight into the average light conditions primary producers experience in the surface layers. Average $\Delta\rho$ during July 2012 indicate a weak stratification in the outer German Bight with values mostly below 0.4 kg m$^{-3}$, with the exception of a small patch south of Helgoland (Fig. 11). During July 2013, $\Delta\rho$ displays an area of strong stratification ($\Delta\rho > 1.0$ kg m$^{-3}$) penetrating to the inner German Bight along the old Elbe Valley. Contributions of temperature and salinity to the $\Delta\rho$, i.e., $\Delta\rho_T$ and $\Delta\rho_S$, as approximately estimated by the linearized equation of state ($\rho - \rho_0 = \alpha(T - T_0) + \beta(S - S_0) + \gamma(P - P_0)$), with $\alpha$ = -0.15 kg m$^{-3}$/K and $\beta$ = 0.78 kg m$^{-3}$ / (g kg$^{-1}$)) suggests that $\Delta\rho_S$ is larger than $\Delta\rho_T$ in a region surrounding and extending northwest of Helgoland. The 2013-R12 scenario results in a $\Delta\rho$ similar in intensity and shape to that in 2013, only narrower in the inner German Bight, whereas the $\Delta\rho$ estimated by the 2013-M12 and 2013-W12 scenarios are small within the outer areas as in 2012, but forms a large patch located northeast of Helgoland.

Simulated DISi and DIN plumes of Elbe in 2013 July following the flood event (Fig. 12) resemble the freshwater plumes (Fig. 10). This plume disappears when the river forcing of 2012 is used (2013-R12) and it gets pushed along the eastern coast in the 2013-M12 scenario (Fig. 12), similar to the freshwater plume (Fig. 10). The plume of DIP on the other hand, when scaled to the Redfield proportions (molar N:P=16), is confined to a smaller region closer to the Elbe estuary. Thus, the impact of the river plume on the nutrients can be tracked by the enhanced N:P ratios.

Spatial distribution of the water-column integrated net primary production rate, NPPR is considerably different in July 2013 than in July 2012 (Fig. 12). Two areas with prominent changes can be distinguished: *i)* outer German Bight (OGB), i.e., west of 7.5 °E and north of 54.5 °N; *ii)* central German Bight (CGB), i.e., region around Helgoland, and its westward and northward extensions. Within the OGB, NPPR estimated for 2013 is lower than 2012 and by 2013-M12/W12. This can be explained by the nutrient limited phytoplankton growth in this region, and the intensification of nutrient limitation due to stronger stratification in 2013 driven by meteorological conditions (Fig. 11). Within the CGB, the distinctive patch of high NPPR that is narrowly present in July 2012 expands considerably in July 2013. In comparison to 2013, the 2013-R12 scenario results in a weakening of NPPR within the entire CGB, in terms of both peak rates and areal coverage of high values, especially in the northern portion. The 2013-M12/W12 scenarios also lead to local reductions of the peak rates achieved at around and north of Helgoland, pointing to the relevance of the hydrological conditions for the intensity of NPPR during July 2013. The enhancement of NPPR within the CGB can be explained by the enhancement of light conditions due to strong stratification in this nutrient-rich region, especially following the flood event (Fig. 12).

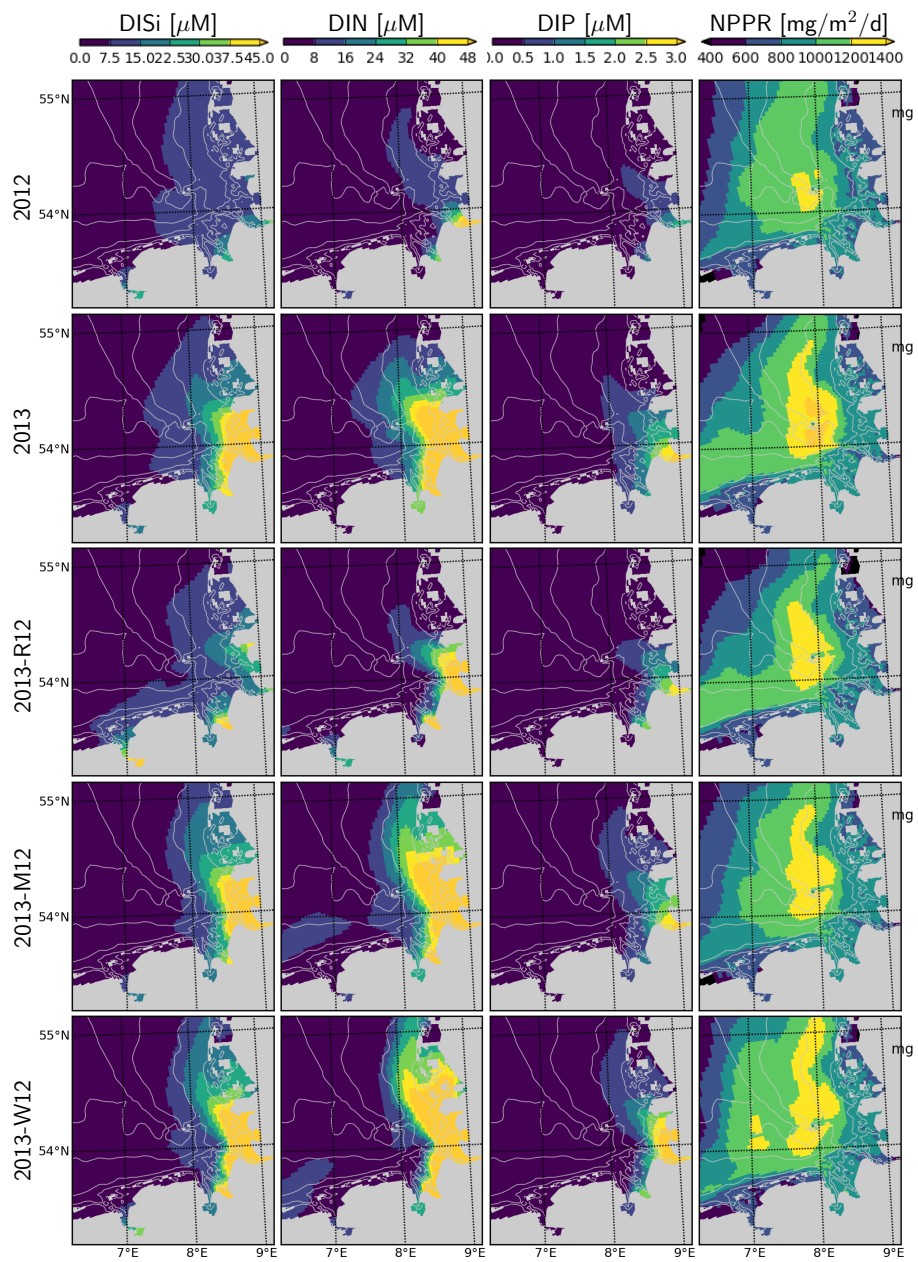

**Figure 12.** Surface DISi, DIN, DIP and integrated net primary production rate during July for the years 2012 and 2013 and scenarios 2013-R12, 2013-M12 and 2013-W12.

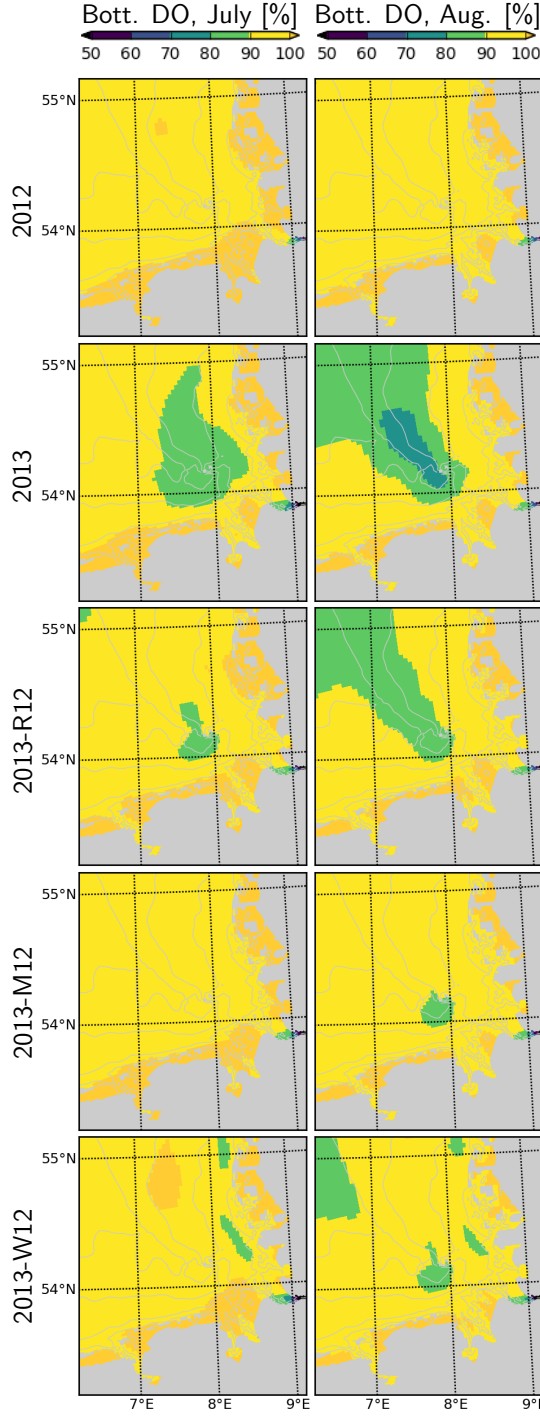

**Figure 13.** Dissolved oxygen in bottom layers during July and August, for the years 2012 and 2013 and scenarios 2013-R12, 2013-M12 and 2013-W12.

In 2012, Dissolved Oxygen (DO) remains close to saturation (Fig. 13). In contrast, in July 2013, a widespread patch of oxygen undersaturation ($< 90\%$ of saturation) develops within the bottom layers of the CGB. This further intensifies ($< 80\%$) and expands towards the OGB during August 2013. Occurrence of this oxygen undersaturation can be explained by the enhanced DO consumption, fueled by the increased NPPR within the CGB (Fig. 12) and the intense stratification within the entire German Bight (Fig. 11) that limits the oxygenation of the bottom layers. In the OGB, the widespread oxygen undersat-

uration despite the lower NPPR (Fig. 12), highlights the importance of stratification (Fig. 11). Under the 2013-R12 scenario, oxygen levels do not drop as much as in the 2013 scenario within the CGB, and the area with oxygen undersaturation shrinks especially during July, but also in August. The 2013-M12 and 2013-W12 scenarios result in a complete disappearance of the oxygen undersaturation within the CGB during July, pointing to the effectiveness of wind-induced mixing in the oxygenation of bottom layers.

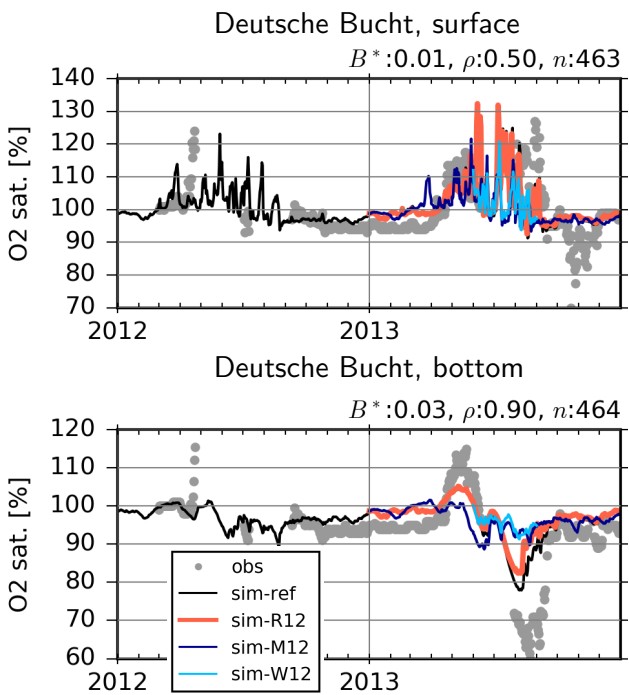

**Figure 14.** Oberved (dots) and simulated (lines) dissolved oxygen in the surface and bottom layer at the Deutsche Bucht station. $B*$: normalized bias, $\rho$: correlation coefficient, $n$: number of observation-simulation pairs based on the reference (ref) run.

At the Deutsche Bucht station, where the temperature and salinity measurements were shown to be reasonably reproduced (Fig. 6), the DO measurements are also mostly well reproduced (Fig. 14). Importantly, the higher levels of supersaturation during 2013 in comparison to 2012, driven by higher NPPR (Fig. 12), and the oxygen depletion in the bottom layers in 2013, and the lack thereof in 2012 are qualitatively captured, although the DO depletion in 2013 is not fully reproduced. Especially the 2013-M12/W12, and to a lesser extent, 2013-R12 scenarios result in lower levels of supersaturation at the surface, indicating

lower levels of NPPR (Fig. 12). At the bottom, especially the 2013-M12/W12 scenarios result in the disappearance of the

oxygen drawdown in July 2013, which is driven by both lesser amounts of organic material to degrade as a result of lower NPPR, and the oxygenation of bottom layers via vertical mixing caused by the windy conditions of 2012. The 2013-R12 results in a lower level of drawdown in comparison to the reference (2013) simulation, and an earlier recovery back to the saturation levels.

In order to demonstrate the effects of the thermohaline structure on the current velocities, we consider two specific days characterized by different wind regimes in June 2013, and compare the original estimates with those obtained with 2013-R12 scenario (Fig.15). In order to remove the movements caused by lunar (M2) tides, the current velocities with 30 min. resolution were averaged over 25h intervals, centered around 12:00 of each day. Differences between the two simulations (Fig.15b,e) reveal an increase in current velocities at the surface within the zone affected by the river plume. In the bottom
layers, differences occur as well, but these are smaller in magnitude (not shown).

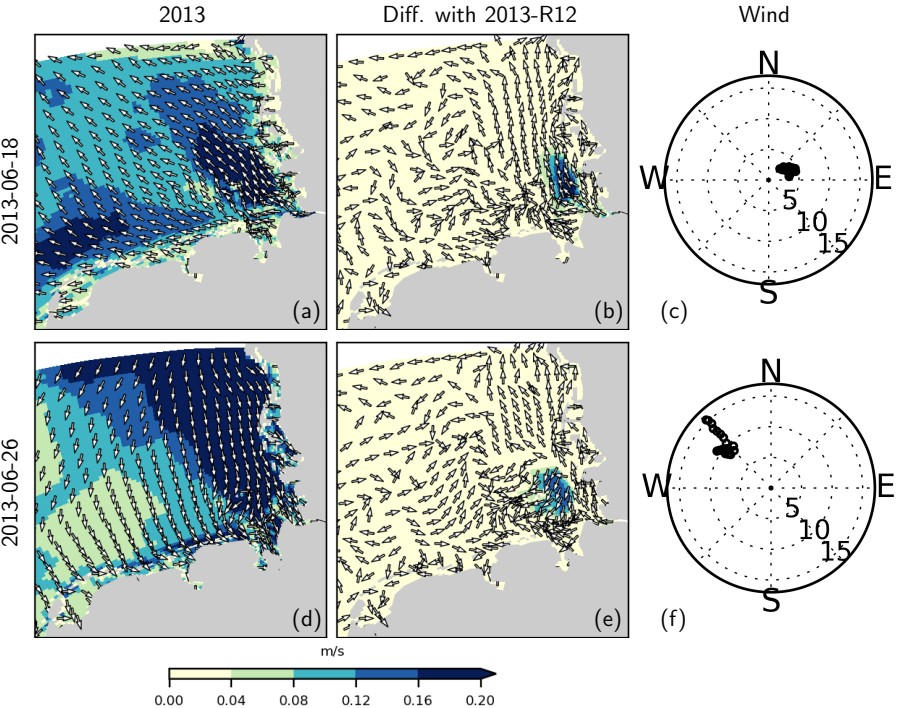

**Figure 15.** 25h-averaged (residual) current velocities at the surface (a,d) and the difference with those obtained with 2013-R12 during two different wind conditions (c,f). In c,f, wind speed at each hour is marked, with distance from origin indicating wind speed, in m/s.

For a better understanding of the modulation of the flow structure by the flood event within the coastal zone, we elaborate 3 cross-shore transects, two of which cross through the monitoring stations, nutrient concentrations at which were displayed in Fig. 9). We focus on the conditions during 18 June 2013 that was considered in Fig. 15a-c, characterized by low wind speeds. On this particular day, an estuarine-like circulation is strongly manifested along the southern part of the north-Frisian
Wadden Sea (see Fig. 1), with the cross-shore (x-) velocities at the bottom layers directed towards the shore, and at the surface

directed off the shore (Fig.16). Removal of the flood event, as predicted by the 2013-R12, results in a weakening of the bottom currents at the southern section (as represented by Suederpiep) and the middle section (as represented by Westerhever). The along-shore (y-) velocities in the bottom layers, directed towards south (outwards the plane) display a similar weakening of the bottom currents. These results provide evidence for the determination of the efficiency of estuarine circulation by an interplay
between the meteorological and hydrological conditions, which are subject to spatio-temporal variations.

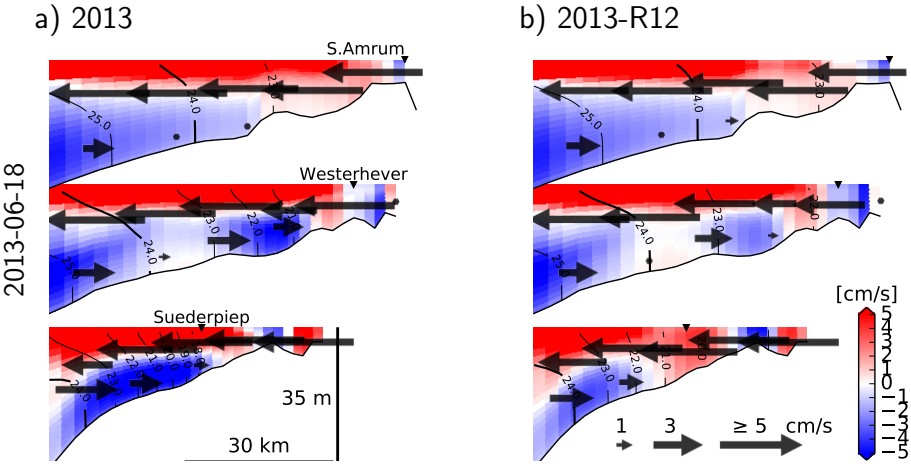

**Figure 16.** 25h-averaged velocity and density structure simulated with the reference model (a) and with the riverine forcing of 2012 (b), under the northeasterly winds on June 18, 2013 (see Fig.15). Two of the transects cross from the stations shown in Fig.9 (marked by ▼ symbols). Arrows indicate the cross-shore velocities, and the colors indicate along-shore velocities with positive values indicating northward flows (i.e., inward the drawing plane). Contour lines indicate $\sigma_T$.

## 4   Discussion

### 4.1   Model Performance

In comparison to the performance of the previous version of the hydrodynamical model setup presented by Kerimoglu et al. (2017a), the ability of the model in representing the cross-shore salinity gradients has been significantly improved, mainly due
to the introduction of flow-dependent horizontal diffusion (e.g., Fig.A1). As suggested by the comparisons with ICES data (Fig. 5), realism of temperature has also been improved, with the normalized bias decreasing from -0.11 to -0.03, and the correlation increasing from 0.95 to 0.99 (compare with Fig. 4 of Kerimoglu et al. (2017a)). There have been incremental improvements in the prediction of nutrient concentrations as well. However, these minor deviations may be related to the differences in specific time periods of interest (2006-2010 in the former study vs 2011-2014 in this study).
The underestimation of salinities (Fig. 7), and consequently the overestimation of nutrient concentrations along the coast (Fig.9) are possibly due to underestimating the flushing rate at the coastal zone. The insufficient spread of coastal waters is

potentially the reason for overestimated salinities and underestimated $NO_3$ in the transition zone, characterized by intermediate salinities and $NO_3$ concentrations (Fig. 5). These errors, in turn, may have lead to an over- and underestimation of the importance of riverine discharges on the stratification dynamics and productivity in the coastal and transition zones, respectively. Before the application of explicit horizontal diffusion, these errors were much larger (Appendix A). Application of higher horizontal diffusion rates (e.g., via higher Smagorinsky coefficient $C_S$, see Appendix A) further improved the model performance along the east Frisian Wadden Sea. However, this was at the cost of overestimation of salinities at the mouth of the estuary, such as at the Cuxhaven station (Fig.6), as well as further dampening of the tidal amplitudes, which were already slightly underestimated (not shown). A spatially variable $C_S$ field, with gradually decreasing values at the mouth of the Elbe helped circumventing this problem, but this spatially variable parameterization was not adopted in this study. Before resorting to such ad-hoc solutions, other potential sources of error need to be assessed.

Despite the potential imperfections in the representation of hydrodynamical processes, the model was able to reproduce various characteristic features of the system, as indicated by the low bias and high correlation coefficients for temperature, salinity and nutrients (e.g., Figs. 5, 6, 8). The skill of the model in reproducing chlorophyll concentrations was not as good (Fig. 8, see below for a discussion of potential reasons). Importantly, the influence of the meteorological and hydrological peculiarities on the hydrodynamical (Figs. 6, 7, A1) and biogeochemical structure of the system were captured (Figs.8, 9, 14).

The skill of the model at the Helgoland station, both with respect to the physical (Fig.6) and biogeochemical variables (Fig.8) is noteworthy, given the heterogeneities caused by the complex topography, and the sharp gradients around the island (Callies and Scharfe, 2015), owed to its location at a coastal transition zone. For instance, the sharp DIN peak observed and simulated at Helgoland during June/July 2013 is uncommon for the summer season (see Fig. 12 in Voynova et al., 2017). Overlapping DIN and freshwater fronts simulated by the model, temporarily spreading to the west of Helgoland during the same period (not shown), and supported also by a sharp decline of observed and simulated salinities (Fig. 6), reveal that this rare summer DIN peak was caused by the plume of the Elbe-Weser flood. This provides evidence for the model's ability to reproduce the behavior of the plume.

## 4.2 Physical and biogeochemical structure of the system

Based on a plethora of *in-situ* observations, Voynova et al. (2017) reported a number of anomalies in the German Bight, following the historical flood event in June 2013, during which, a large quantity of freshwater and nutrients were delivered to the coast by the Elbe and Weser rivers within a short time period (Fig. 3). Our numerical simulations are in agreement with many of those findings, such as the anomalous spatial distribution of salinity, nitrogen and silicate following the flood event (e.g., compare Fig. 10, 12 with the Fig. 11 of Voynova et al. (2017)).

In addition, our findings point to the relevance of the meteorological conditions that interact with the impacts of the flood event. In particular, our findings suggest that mainly the wind conditions (Fig. 4) resulted in a particularly intense stratification (Fig.11). Within the central German Bight, a combination of thermal and haline dynamics extended the area of intense stratification. The thermohaline dynamics in the inner German Bight have been recognized before (Frey, 1990; van Leeuwen et al., 2015). Following the flood event, these interactions have moved away from the coast to further offshore regions of the German

Bight. It should be noted that, variations in stratification intensity driven by the spring and neap tides as in the Rhine ROFI (Simpson et al., 1993) have been identified for our study system as well, but these are relevant at shorter (weekly) time scales (Chegini et al., submitted).

The enhanced water column stability (Fig. 11), and hence reduced light limitation, in combination with higher nutrient availability supplied by the flood event (Fig. 3, 12), increased the NPPR within the central German Bight (Fig. 12), which may explain the high pH and DO oversaturation reported by Voynova et al. (2017). In turn, the combination of uninterrupted phases of stratification during July that gave rise to a large average density difference (Fig. 11) and the breakdown of high amounts of organic material as a result of enhanced NPPR (Fig. 12), following the flood event, lead to a widespread oxygen depletion in the bottom layers. The DO supersaturation in the surface layers, and subsequent undersaturation in bottom waters observed in the Deutsche Bucht station, which was previously documented by Voynova et al. (2017), was correctly captured by the model (Fig.14). The scenario analysis suggests that, especially the meteorological conditions during the summer of 2013, but also the flood event were relevant for the occurrence and intensity of this oxygen drawdown in the German Bight (Fig. 13-14). This explains why such a degree of oxygen depletion in the German Bight is unusual (e.g., Voynova et al., 2017; Große et al., 2016, 2017). Within the outer German Bight, the higher water column stability lead to an intensification of the nutrient limitation within the upper mixed layer, and consequently lower NPPR (Fig. 12). At the vicinity of the mouths of the Elbe and Weser rivers, NPPR did not respond strongly to the flood (Fig. 12), as these areas were limited by light, rather than nutrients (see also Loebl et al., 2009). In reality, an even stronger light limitation in the vicinity of the mouth of the Elbe estuary is likely, due to the increase in the SPM towards the Elbe estuary (van Beusekom and Brockmann, 1998; Gayer et al., 2006), which is only partially accounted for by the model (see Appendix B2). It should be noted that the riverine influence within the coastal zone may be overestimated by our simulations, given the lower than observed salinities (Fig. 7), and higher than observed nutrient concentrations (Fig. 9).

Our results point to an increase in current velocities at the surface under the influence of the 2013 flood (Fig. 15), which is presumably driven by the reduced dissipation of kinetic energy through vertical mixing, owed to the intensification of haline stratification (Fig. 11), i.e., the baroclinicity of the current structure. Enhancement of the current velocities at the surface, in turn, might have facilitated the spread of the plume towards the outer German Bight in 2013 (Fig.10-12). However, the main reason for the eastward spread of the plume is the wind conditions, which presumably lead to a dominance of anticyclonic circulation during July 2013, as was also suggested by a principal component analysis of a barotropic model simulation (https://coastmap.hzg.de/coastmap/modeldata/model1/#/residualcurrents, see Callies et al. (2016) for data access). It has been shown that the residual surface currents in the German Bight are largely determined by the wind patterns (Schrum, 1997; Callies et al., 2017).

van Beusekom et al. (2019) has shown and discussed the presence of regional differences in thermohaline estuary-type circulation (as in Burchard and Badewien, 2015; Hofmeister et al., 2017) in the Wadden Sea. Here, our results suggest that the strength of the thermohaline estuarine circulation (Burchard and Badewien, 2015) can be enhanced by surplus buoyancy fluxes, here driven by the flood event (Fig. 16). This is as expected, and can enhance coastal accumulation of SPM and nutrients even away from the estuary itself (Hofmeister et al., 2017).

Our model-based analysis here is not conclusive, but rather exploratory. Given the anticipated increase in the frequency and intensity of the hydro-meteorological extremes due to climate change (Beniston et al., 2007; Wetz and Yoskowitz, 2013), further research is needed to understand the processes underlying the interactive impacts of these events on the physical and biogeochemical structure of the coastal systems and estuaries. Such a mechanistic understanding is essential for policy making, such as the regulation of nutrient loading rates in rivers (see, e.g., OSPAR, 2017).

### 4.3 Model limitations and perspectives

Since the first 3D models of the North Sea (Backhaus, 1985; Dippner, 1993; Schrum, 1997), computational capacity has been significantly improved, which resulted in development of ever finer resolution setups that can resolve meso-scale features such as the coastal freshwater fronts and baroclinic eddies (Holt and James, 2006; Pohlmann, 2006; Staneva et al., 2009; Pätsch et al., 2017), and smaller-scale dynamics, such as the estuarine processes (Gräwe et al., 2016; Stanev et al., 2019; Pein et al., 2019). For large-domain biogeochemical applications that require a costly calculation of transport of many additional state variables, the coarse-resolution models (10-20 km) are being actively used (e.g., Große et al., 2016; Ford et al., 2017; Daewel et al., 2019). With a spatial resolution of 1.5-4.5 km covering the southern North Sea (Fig. 1), the setup we employed here falls in the middle of the spectrum, and is similar to the setup used by Los et al. (2008) and the 'southeastern North Sea' setup of Androsov et al. (2019).

A potential source of bias in salinity and nutrients along the Elbe-plume is the misrepresentation of the Elbe estuary in our model setup (Fig. 1). For instance, according to a recent, high resolution model of the Elbe estuary, the freshwater-saline water transition (0-5 g/kg) occurs at about 50-75 km upstream of the mouth of Elbe (under normal hydrological conditions), and the N and Si concentrations vary considerably within the estuary (Pein et al., 2019). Indeed, a high resolution (300m) setup of the German Bight that resolves up to 150 km upstream the Elbe mouth (Chegini et al., submitted) demonstrated better skill in reproducing the salinity observations shown in Fig.7 and Fig. A1. Another contingent error source is potential inaccuracies in advective transport rates, e.g., as a result of imperfections in meteorological forcing (Geyer, 2014); or ignoring the effects of off-shore wind-farms on the thermohaline circulation (Carpenter et al., 2016; Platis et al., 2018). In order to assess the realism of the advective transport rates estimated by our hydrodynamic setup, we are planning to do a comparison with other models, such as the operational model of the BSH (see Callies et al., 2017).

The structure and process descriptions used for the biogeochemical model introduced in this study are similar to those used recently for studying the interaction between the hydrodynamical and biogeochemical processes in coastal systems, in particular, nutrient cycling and oxygen dynamics, in the North Sea (e.g. Große et al., 2016; Kerimoglu et al., 2017a), the Elbe estuary (Pein et al., 2019) and other similarly dynamic coastal shelf systems, such as the Louisiana Shelf (Fennel and Laurent, 2018) and the Chesapeake Bay (Irby et al., 2018). Description of the non-planktonic components, consisting of two detritus classes, dissolved organic material, dissolved inorganic nutrients, oxygen, and a simple benthic pool to represent benthic remineralization, oxygen consumption and denitrification (Fig. 2) were largely based on ECOHAM (see Section 2.1 Appendix B), which was earlier derived from ERSEM (Baretta et al., 1995). Unlike ECOHAM, but like in a majority of the aforementioned models (Feng et al., 2015; Kerimoglu et al., 2017a; Laurent et al., 2017; Pein et al., 2019), DOM remineralization is described

as first order kinetics, instead of mediated by an explicitly described bacteria, which, considering the purposes of the model, we consider to be non-critical.

Water column processes dominate the organic matter turnover in coastal systems amounting to about 80% in the deeper parts of the German Bight to around 50% in the Wadden Sea (Heip et al., 1995; van Beusekom et al., 1999). Based on these estimates, within the outer German Bight, we do not expect a large sensitivity of our results to the resolution of benthic processes, but this may be the case for the shallower sites and the Wadden Sea. Underestimated oxygen depletion (Fig. 14), and the inability of the model to capture some high P concentrations (reflected as sporadic, but large underestimation errors in Fig. 5), are possibly related to the oversimplifications in the benthic model. The simulated benthic oxygen consumption rates, of up to 15 mmol m$^{-2}$ d$^{-1}$ during the summer months are less than half of the upper range of measurements in the German Bight (e.g. Ahmerkamp et al., 2017; Neumann et al., 2017). Since the benthic oxygen consumption rates are calculated based on the benthic remineralization rates in our model, the underestimation of benthic oxygen consumpion implies an underestimation of the role played by the sediments in nutrient cycling in some locations and times of the year. The foremost reason for this underestimation is likely the inaccuracies in POM sedimentation flux as determined as a fraction of sinking rates of detritus in our model (Table B8). Sedimentation fluxes and benthic transformation rates, in reality, are recognized to be controlled by the spatially heterogeneous sediment permeability (Ahmerkamp et al., 2017) and increasingly, by the activity of benthic organisms, which vary not only spatially, but also temporally (e.g., Singer et al., 2016). The micro- and macroalgae, for instance, which can form substantial biomass in the intertidal mudflats of the Wadden Sea (Christianen et al., 2017), may contribute to the imbalances in the cycling rates of C, N and P (Cook et al., 2007), and due to producing high amounts of extracellular polymeric substances, alter the resuspension and sedimentation rates (Hanlon et al., 2006). Bioturbating animals alter the decomposition and oxygen consumption rates, directly, by their own consumption (Middelburg, 2018), and indirectly, by altering the solute transport and ventilation rates (Herman et al., 1999). Suspension feeders and sponge, on the other hand, by consuming the suspended particulate matter and DOC in the water column and excreting into the sediments, enhance the water-to-sediment flux (Middelburg, 2018). The response of benthic macrofaunal groups to changes in pelagic productivity (Fig. 12) and the deterioration of the oxygenation state (Fig. 13) caused by the flood event may differ (Rosenberg et al., 2002; Lessin et al., 2019). This may imply alterations of the benthic functioning at time scales that may well exceed the scales studied here. Analyses of such effects with models require a very detailed representation of the benthic communities and their function. Such models are rare (but see, e.g., Baird et al., 2016; Lessin et al., 2019), however, this is expected to change with the improving data availability and computational capacities (Lessin et al., 2018). An intermediate step might be to use statistically estimated distribution of benthic organisms for various scenarios (e.g., Singer et al., 2017) as external forcing in biogeochemical models (see, e.g., Nasermoaddeli et al., 2018).

In spite of its simplicity, the benthic model is partially successful in estimating the denitrification rates: for the German Bight, the simulated rates reach to about 1.5 mmol N m$^{-2}$ d$^{-1}$ during summer months, which is close to the upper measurement range of 1.9 (e.g., Bratek et al., 2020). Adjacent to the Elbe Estuary, the model reproduces the strong heterogeneity in denitrification rates (Deek et al., 2013), and to a satisfactory degree, the measured rates as well: at the 'NW' station of Deek et al. (2013, their Fig.1), the simulated denitrification of 2.5-3 mmol N m$^{-2}$ d$^{-1}$ for 2012, only slightly overestimates the measured rates

of approximately 2.2-2.6 mmol N m$^{-2}$ d$^{-1}$ (Deek et al., 2013, their Fig.3). Following the flood event in 2013, simulated benthic denitrification rates within the areas adjacent to the Elbe Estuary increase by more than 50%. This increase is solely driven by the increased POM loading during the flood (Fig. 3) according to our model. But in reality, the enhanced oxygen depletion in bottom layers (Fig.13) may lead to a disproportional benthic deoxygenation, which may, in turn, enhance benthic denitrification rates within the off-shore regions as well.

The lack of an explicit representation of the benthic oxygen profiles and redox reactions may have contributed to the underestimation of benthic oxygen consumption as well, although it was shown that a vertically integrated approximation like the benthic model we used, especially when combined with meta-model parameterizations, can reliably behave like a computationally demanding, vertically resolved explicit diagenetic model (Soetaert et al., 2000). The sporadic large underestimation errors in P concentrations are identified to occur in some coastal regions during summer months, when the nitrogen concen-
trations are at their lowest. Such decoupling of phosphorus and nitrogen in certain Wadden Sea regions is well known, and recognized to be driven by the depletion of benthic oxygen during summer, which leads to release of iron-bound P, while promoting denitrification in the sediments (see, e.g. Loebl et al., 2007; Grunwald et al., 2010; Leote et al., 2015). Although the latter is accounted for by our model (Table B8), the former is not, which can explain the inability of the model in capturing the late-summer P peaks.

In three out of four stations we considered, chlorophyll concentrations are overestimated (Fig. 8). Considering these biases, rather than the absolute values of NPPR estimates, simulated responses to hydro-meteorological forcing should be regarded (Fig. 12). Reasons for the overestimation of chlorophyll concentrations seem to be region specific: overestimation of winter concentrations in Terschelling-50 and Noordwijk-70 suggest insufficient respiration rates, whereas spring blooms starting too early at Helgoland suggest inaccuracies in the seasonality of under-water light climate. During the summer months, misrepre-
sentation of grazing losses and vertical distribution of chlorophyll (e.g., van Leeuwen et al., 2013; Kerimoglu et al., 2017a) may have contributed to the overestimation errors as well. A detailed identification of the chlorophyll dynamics therefore require a careful consideration of all these factors and comparisons against additional datasets, which is outside of the scope of this study. However, differences in baseline concentrations at different stations during summer are quite realistically reproduced, suggesting that the large-scale gradients are realistically represented (Fig. 8), which we consider to be sufficient for the purposes of this
study. The structure of the plankton food web assumed in this study, consisting of two phytoplankton (flagellates and diatoms) and two zooplankton (micro- and mesozooplankton) groups, is similar to those by Große et al. (2017) and Pein et al. (2019), but here the variability in phytoplankton cellular composition were taken into account, using the Droop and Geider et al. (1997) formulations to resolve the variability in C:N:P and Chl:C, respectively, similar to as in, e.g., ERSEM (e.g., Ford et al., 2017). In the future, we are planning to improve the representation of other plankton groups in the system, such as colony-forming
*Phaeocystis* and mixotrophic forms, which can be abundantly found in the coastal waters of the southern North Sea (Löder et al., 2012; Burson et al., 2016). A module that provides a simplistic description of mixotrophy (as in Kerimoglu et al., 2017b), is already available, but we chose not to use it in this study, for the sake of avoiding increasing the model complexity further.

Given the limitations of the biogeochemical model discussed above, its predictions should not be interpreted in an absolute sense. However, the simulated responses by the model are plausible, and therefore the analysis presented in this study is

490 expected to be of heuristic value in gaining a systematic understanding of the role of riverine and meteorological forcing in shaping of the hydrodynamical and biogeochemical structure of the system.

## 5    Conclusions

In this study, we presented a newly developed biogeochemical model and improvements of a hydrodynamical model described in an earlier study. The coupled hydrodynamical-biogeochemical model system is shown to satisfactorily reproduce the char-
495 acteristic features of the German Bight ecosystem, and the impacts of a 100-year flooding of the Elbe and Weser rivers. Our results reveal that the flood event coincided with special meteorological conditions in the region, namely a calm and warm summer dominated by an anticyclonic circulation, resulting in a particularly intense and widespread stratification. The stronger stratification, and the increased availability of nutrients impacted the primary production in the system and the oxygen levels in the bottom waters. Through a scenario analysis, we found that the observed anomalies in July 2013 were likely driven by the
500 meteorological conditions within the outer German Bight, and the interaction between meteorological and hydrological conditions within the central German Bight, suggesting that the impacts of flood events in the system are context-dependent. These extreme flooding and meteorological conditions may occur more frequently in the future, which requires a better understanding of the mechanisms governing the response of the coastal systems to such extreme events.

*Code and data availability*.  Codes of the hydrodynamical models and the coupler are available at the following git repositories: GETM:
https://sourceforge.net/p/getm, GOTM: https://github.com/gotm-model, FABM: https://github.com/fabm-model/fabm. The biogeochemical model code will be released in the near future, but a beta version can be provided by OK upon request. ICES and COSYNA data used for model validation are available from https://ices.dk and https://cosyna.de, respectively. Data from Terschelling and Noordwijk stations are available at https://waterinfo.rws.nl. Surface elevation, meteorological and EMEP atmospheric deposition data used as model forcing are available from, respectively: https://doi.org/10.1594/WDCC/coastDat-2_TRIM-NP-2d, https://doi.org/10.1594/WDCC/coastDat-2_
COSMO-CLM and https://emep.int. EMODnet bathymetry data is available from https://emodnet-bathymetry.eu. Model output of the current study will be provided by OK upon request.

## Appendix A: Description of horizontal diffusion in the hydrodynamical model

Modern advection schemes (including TVD-transport as used in many coastal applications) are developed and tested for homogeneous grid spacing (Pietrzak, 1998; Barthel et al., 2012), although coastal applications tend to use varying grid spacing in curvilinear horizontal, unstructured horizontal and general vertical grids (e.g., Zhang et al., 2016; Kerimoglu et al., 2017a). The performance of slope limiters and the involved numerical mixing is therefore almost unpredictable for two reasons: a) tracer mixing is ultimately always a combination of numerical mixing and physical mixing terms, both effects reduce each other (Hofmeister et al., 2011), and b) numerical mixing as a nonlinear effect of the advection is seldom analyzed in model applications. Comparisons of the mixing term strength between model applications then potentially result in differences of the advection scheme performance, more than an analysis of the physical effect of mixing mass concentrations.

There exists a plethora of methods for the specification of horizontal diffusion or isopycnal mixing for ocean models (see, e.g. Gent and McWilliams, 1990; Roberts and Marshall, 1998; Beckers et al., 2000; Griffies and Hallberg, 2000), a review or discussion of which is beyond the scope of this appendix. Here, we will demonstrate the use of a simple subgridscale parameterization by Smagorinsky (1963), which was originally for modelling atmospheric circulation, and is now commonly used in both atmospheric and ocean circulation models (Becker and Burkhardt, 2007). The magnitude of horizontal diffusivity is recognized to exhibit strong variations in space and time (Wang, 2003). The Smagorinsky parameterization achieves such variations by scaling the diffusion coefficient proportionally with the grid size and deformation rates of lateral velocities, e.g., for the horizontal diffusion of momentum:

$$A_M = C_S * \Delta x \Delta y * \sqrt{\left(\frac{\partial u}{\partial x}\right)^2 + \left(\frac{\partial v}{\partial y}\right)^2 + \frac{1}{2}\left(\frac{\partial u}{\partial y} + \frac{\partial v}{\partial x}\right)^2} \tag{A1}$$

where, $C_S$ is the empirical Smagorinsky constant, $u$, $v$, $\Delta x$ and $\Delta y$ are velocities and grid spacings along $x$ and $y$ dimensions, respectively. Then the horizontal diffusion of tracers, $A_H$ follows:

$$A_H = A_M/Prt \tag{A2}$$

In (A1), $C_S$ is not physically well constraint, but is adjusted based on numerical considerations (Kantha and Clayson, 2000), e.g., the diffusion vs. dispersion trade-off (Pietrzak, 1998). In this study, we set $C_S = 0.6$ and Prandtl number, $Prt = 1.0$, and examine the effects of this parameterization on the representation of the river plume during 2012-2013, with a focus on the freshwater plume during the flooding event. Specifically, we compare the predictions of 2 model variants against the Ferrybox measurements taken by the platform installed on the M/V *Funny Girl* ferry, that are analyzed in greater detail in the main text (Fig. 7).

The variant where no diffusion was enabled, overestimates the cross-shore salinity gradient along the North Frisian coast, i.e., North of Elbe, in the form of too low near-coast salinities (Fig.A1b). On the other hand, the variant where horizontal diffusion was described with Smagorinsky parameterization, have considerably better skill in reproducing the FerryBox measurements along the Büsum-Helgoland ferry track (Fig. A1c).

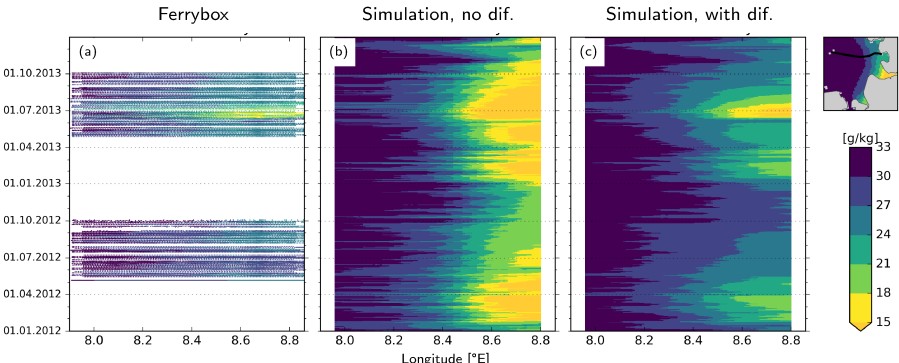

**Figure A1.** Hovmöller diagrams of salinity distribution in 2012 and 2013 along the (average) transect shown at the top right corner, as measured by the FerryBox platform (a) and models without (b) and with (c) horizontal diffusion.

Plausibility of the total horizontal mixing, and its physical and numerical components can be diagnosed by an analysis of the discrete variance decay (DVD) of salinity (Klingbeil et al., 2014) based on Burchard and Rennau (2008). In the absence of
545 explicit diffusion, the sum of physical and numerical mixing becomes negative at the mouth of the Elbe and Weser rivers, and within their ROFI, implying spuriously enhanced horizontal gradients (Fig. A2c). With the application of explicit diffusion, numerical mixing values effectively decrease both within the positive and negative spectra (Fig. A2d), leading to near-complete elimination of negative values in total mixing (Fig. A2f).

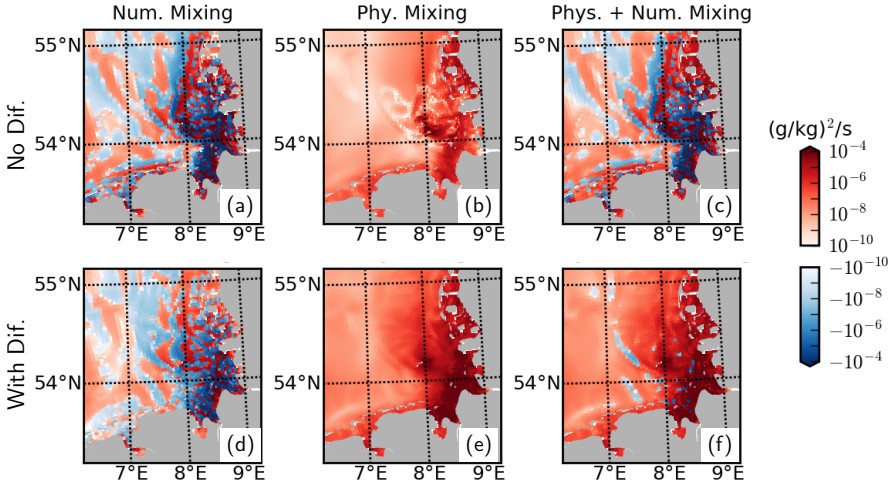

**Figure A2.** Mixing analysis for July 2013 based on temporally averaged values at the surface.

We conclude that application of explicit horizontal mixing through a simple parameterization can be useful in improving
the skill of a 3-D coupled physical-biogeochemical model within the vicinity of river discharges, and eliminate implausible negative total (physical + numerical) mixing values.

## Appendix B: Detailed description of the biogeochemical model

All modelled state variables and fluxes between various pools are shown in Fig. 2. In the following sections, sink and source terms for the planktonic and abiotic variables ($s(v)$ in Tables B1, B7) and the description of processes (Tables B2, B4, B9) will be provided. For describing the fluxes between various pools, where possible, we adopt the $source\_target$ notation as in Pätsch and Kühn (2008), which was earlier adopted from ERSEM (Blackford and Radford, 1995). Although this notation is consistent with that used in the Fortran module of abiotic components, the programming notation of the plankton module is somewhat different, due to their different historical origins.

All kinetic rates in planktonic and abiotic components are modified with temperature using the Q10 rule:

$$f_{T,(phy,zoo-j,abio)} = Q10_{(phy,zoo-j,abio)}^{(T-T_{ref})/T_{ref}} \tag{B1}$$

with $T_{ref} = 10°C$, and $Q10_{phy} = Q10_{zoo-mic} = Q10_{abio} = 1.5$ and $Q10_{zoo-mes} = 2.0$.

### B1 Planktonic components

The plankton model was developed based on Kerimoglu et al. (2017b) regarding the modularity concept that allows coupling plankton units in run time (see Bruggeman and Bolding, 2014), as well as description of internal variation of P quota of phytoplankton (B2,B4,B12) according to the Droop model (as in Morel, 1987). Here, we further considered the uptake $NO_3$ and $NH_4$ of phytoplankton (similar to Pätsch and Kühn, 2008), and the resulting variations of N quota (B3,B13); limitation of diatoms by Si (B11) using a Monod-type relationship (Flynn, 2003); dependence of the light limitation on the chlorophyll content, i.e., $\theta$ (B15), in phytoplankton (B9), and the dynamic variations of $\theta$ (B5,B16) following Geider et al. (1997). The plankton module provides various options for the representation of nutrient and carbon limitation in a consistent way, which is intended to be further enriched and elaborated in future studies. Given the suppleness of the model with respect to the description of physiological processes and interactions between plankton groups, the model is provisionally named as the 'Generalized Plankton Model' (GPM).

**Table B1.** Source-sink terms of the dynamic variables (all in mmol/m$^3$/d) of the plankton module. Indices $p_i$= {diatoms, flagellates}; $z_j$={microzooplankton, mesozooplankton}; $t_k$= zooplankton target.

| | | | |
|---|---|---|---|
| C bound to $p_i$ | $s(p_i^C)$ | $= DIC\_p_i^C - p_i^C\_DOC - M_i^C - \sum_j I_{j,i}^C \cdot z_j^C$ | (B2) |
| N bound to $p_i$ | $s(p_i^N)$ | $= NO3\_p_i^N + NH4\_p_i^N - p_i^N\_DON - M_i^N - \sum_j I_{j,i}^N z_j^C$ | (B3) |
| P bound to $p_i$ | $s(p_i^P)$ | $= DIP\_p_i^P - p_i^P\_DOP - M_i^P - \sum_j I_{j,i}^P \cdot z_j^C$ | (B4) |
| Chl bound to $p_i$ | $s(p_i^{chl})$ | $= \rho_i \cdot DIC\_p_i^C - (p_i^C\_DOC + M_i^C + \sum_j I_{j,i}^C \cdot \theta_i \cdot 12.0[gC/molC] \cdot z_j^C$ | (B5) |
| C bound to $z_j$ | $s(z_j^C)$ | $= \sum_k t_k^C\_z_j^C - zC_j\_DIC - M_j^C$ | (B6) |

As in Kerimoglu et al. (2017a), sinking rate of phytoplankton is formulated as a function of their nutrient status.

$$w_{p,i} = w'_p \cdot \left( 0.1 + 0.9 \cdot \exp\left( -5.0 * \min\left( \frac{QP_i - QP_{min,i}}{QP_{max,i} - QP_{min,i}}, \frac{QN_i - QN_{min,i}}{QN_{max,i} - QN_{min,i}} \right) \right) \right) \tag{B7}$$

In (B7) and in Tables B1-B2, $QX = X:C$ within a certain phytoplankton or zooplankton pool, which may be either a fixed constant (as provided in Table B3), or diagnostically calculated from the instantaneous values (for $X = P, N$ quota of phytoplankton). Exudates of the phytoplankton are assumed to be in DOM form (B2,B17).

**Table B2.** Process equations and functional relationships used in the phytoplankton module

| | | | |
|---|---|---|---|
| C uptake rate of $p_i$ | $DIC\_p_i^C$ | $= p_i^C \cdot f_{T,phy} \cdot V_{max,i}^C \cdot f_{I,i} \cdot \min(f_{N,i}, f_{P,i}, f_{Si,i})$ | (B8) |
| Light limitation of $p_i$ | $f_{I,i}$ | $= 1.0 - \exp\left( \frac{-\alpha_i \cdot \theta_i \cdot I}{f_{T,phy} \cdot V_{max,i}^C \cdot \min(f_{N,i}, f_{P,i})} \right)$ | (B9) |
| Nutrient (X={N,P}) limitation of $p_i$ | $f_{X,i}$ | $= 1.0 - QX_{min,i}/QX_i$ | (B10) |
| Silicate limitation of diatoms | $f_{Si,i}$ | if $i : diatoms = \frac{DISi}{K_i^{Si} + DISi}$, else $= 1.0$ | (B11) |
| DIP uptake rate of $p_i$ | $DIP\_p_i^P$ | $= p_i^C \cdot f_{T,phy} \cdot V_{max,i}^P \cdot \frac{QP_{max,i} - QP_i}{QP_{max,i} - QP_{min,i}} \cdot \frac{DIP}{K_i^P + DIPi}$ | (B12) |
| DINX (NX={NO$_3$,NH$_4$}) uptake rate of $p_i$ | $DIX\_p_i^N$ | $= p_i^C \cdot f_{T,phy} \cdot V_{max,i}^N \cdot \frac{QN_{max,i} - QN_i}{QN_{max,i} - QN_{min,i}} \cdot \frac{DINX/K_i^{NX}}{1.0 + \sum_X DINX/K_i^{NX}}$ | (B13) |
| Silicate uptake rate of $p_i$ | $DISi\_p_i^{Si}$ | if $i : diatoms = DIC\_p_i^C \cdot QSi_i$, else $= 0.0$ | (B14) |
| Chl:C ratio bound to $p_i$ | $\theta_i$ | $= p_i^{chl}/(p_i^C \cdot 12[\text{gC/molC}])$ | (B15) |
| Ratio of chl. synthesis to C fixation | $\rho_i$ | $= \frac{DIC\_p_i^C/p_i^C}{\alpha_i \cdot \theta_i \cdot I}$ | (B16) |
| X (=C,N,P) exudation of $p_i$ | $p_i^X\_DOX$ | $= DIC\_p_i^C \cdot \gamma_i \cdot QX_i$ | (B17) |
| X (=C,N,P) Mortality rate of $p_i$ | $M_i^X$ | $= p_i^X \cdot f_{T,phy} \cdot (m1_i + p_i^C \cdot m2_i)$ | (B18) |

     Process formulations for the zooplankton module are provided in Table B4. Following Fasham et al. (1990), prescribed preferences of prey items for each zooplankton (Table B6) are dynamically weighed with their relative abundance to determine

the effective preferences (B26,B27). Zooplankton are assumed to excrete into $DIM$ pool (B6,B24). As in Kerimoglu et al. (2017b), assimilated and un-assimilated fractions of the ingested prey by each zooplankton $j$ are determined by the assimilation efficiency $\epsilon_j^X$ (B28),B29), which is continuously adjusted (as in Grover, 2002), such that the zooplankton can maintain their homeostatic elemental composition. Here this scheme was extended to multiple nutrients, i.e., N and P, and $\epsilon^X$ are calculated iteratively, similar to that in (Kerimoglu et al., 2018). Starting from each $\epsilon^X$ set to default values (Table B5), if P to be ingested

would be less than the amount required to match the ingested C, $\epsilon^C$ is down-regulated, and vice versa:

$$\epsilon^P = \frac{\epsilon_j^C \cdot \sum_k I_{j,k}^C \cdot QP_j}{\sum_k I_{j,k}^P}, \quad \text{if } \epsilon_j^P \cdot \sum_k I_{j,k}^P > \epsilon_j^C \cdot \sum_k I_{j,k}^C \cdot QP_j \tag{B19}$$

$$\epsilon^C = \frac{\epsilon_j^P \cdot \sum_k I_{j,k}^P}{\sum_k I_{j,k}^C \cdot QP_j}, \qquad \text{otherwise} \tag{B20}$$

**Table B3.** Parameters of the phytoplankton module. Where necessary, multiple values were provided for diatoms and flagellates. Sources: G98: based on Geider et al. (1998); K17: Kerimoglu et al. (2017b); L12: Lorkowski et al. (2012); A: assumed; C: calibrated.

| Symbol | Description | Value$_i$ | Unit | Source |
|---|---|---|---|---|
| $V_{\max,i}^{C}$ | Maximum C uptake rate | 3.0, 2.0 | d$^{-1}$ | G98 |
| $V_{\max,i}^{N}$ | Maximum N uptake rate | 0.3, 0.6 | molN (mmolC d)$^{-1}$ | G98 |
| $V_{\max,i}^{P}$ | Maximum P uptake rate | 0.01, 0.02 | molP (mmolC d)$^{-1}$ | A |
| $K_i^{NO3}$ | Half saturation constant for NO$_3$ uptake | 3.0 | mmolN m$^{-3}$ | G98 |
| $K_i^{NH4}$ | Half saturation constant for NH$_4$ uptake | 1.0 | mmolN m$^{-3}$ | A |
| $K_i^{P}$ | Half saturation constant for P uptake | 0.4 | mmolP m$^{-3}$ | K17 |
| $K_i^{Si}$ | Half saturation constant for Si limitation | 1.0 | mmolSi m$^{-3}$ | A |
| $QS_{i\,\mathrm{diat}}$ | Fixed Si:C ratio of diatoms | 0.17 | molSi molC$^{-1}$ | L12 |
| $QN_{\max,i}$ | Maximum quota for N | 0.18 | molN molC$^{-1}$ | G98 |
| $QP_{\max,i}$ | Maximum quota for P | 0.008 | molP molC$^{-1}$ | A |
| $QN_{min,i}$ | Subsistence N quota | 0.045, 0.06 | molN molC$^{-1}$ | G98 |
| $QP_{min,i}$ | Subsistence P quota | 0.002, 0.003 | molP molC$^{-1}$ | A |
| $\alpha_i$ | Chl. sp. slope of P-I curve | 9.0, 6.0 | gC gChl$^{-1}$ / (molE m$^{-2}$) | G98 |
| $\theta_{max,i}$ | Max. chl:C ratio | 0.10, 0.07 | m$^2$ gChl/gC$^{-1}$ | A |
| $\gamma_i$ | Exudation fraction | 0.05 | - | L12 |
| $m1_i$ | Linear mortality rate | 0.05 | d$^{-1}$ | C |
| $m2_i$ | Quadratic mortality rate | 0.001 | d$^{-1}$/(mmolC m$^{-3}$) | C |
| $\delta_{S,i}$ | Fraction of dead cells diverted to small det. | 0.7,1.0 | - | L12 |
| $w'_p$ | Maximum potential sinking rate | 4.0, 0.2 | m d$^{-1}$ | C |

Next, following the same logic, $\epsilon^N$ and $\epsilon^C$ are regulated to match the C- and N-intake according to the $QN_j$:

$$\epsilon^N = \frac{\epsilon_j^C \cdot \sum_k I_{j,k}^C \cdot QN_j}{\sum_k I_{j,k}^N}, \quad \text{if } \epsilon_j^N \cdot \sum_k I_{j,k}^N > \epsilon_j^C \cdot \sum_k I_{j,k}^C \cdot QN_j \tag{B21}$$

$$\epsilon^C = \frac{\epsilon_j^N \cdot \sum_k I_{j,k}^N}{\sum_k I_{j,k}^C \cdot QN_j}, \qquad\qquad \text{otherwise} \tag{B22}$$

Finally, $\epsilon^P$ is adjusted again, as a potential modification of $\epsilon^C$ in (B21) may require an updated P-intake:

$$\epsilon^P = \frac{\epsilon_j^C \cdot \sum_k I_{j,k}^C \cdot QP_j}{\sum_k I_{j,k}^P}, \quad \text{if } \epsilon_j^P \cdot \sum_k I_{j,k}^P > \epsilon_j^C \cdot \sum_k I_{j,k}^C \cdot QP_j \tag{B23}$$

**Table B4.** Process equations and functional relationships used in the zooplankton module.

| | | | |
|---|---|---|---|
| X (=C,N,P) Excretion rate of $z_j$ | $z_j^X\_DIX$ | $= z_j^C \cdot e_j \cdot f_{T,zoo-j} \cdot QX_j$ | (B24) |
| X (=C,N,P) Mortality rate of $z_j$ | $M_j^X$ | $= z_j^C \cdot (m1_j + z_j^C \cdot m2_j) \cdot QX_j$ | (B25) |
| Ingestion rate of X from $t_k$ | $I_{j,k}^X$ | $= I_{max,j} \cdot f_{T,zoo-j} \cdot \dfrac{pw_{j,k} \cdot t_k^C}{K_j^C + \sum_k (pw_{j,k} \cdot t_k^C)} \cdot QX_k$ | (B26) |
| Weighed preference of target $k$ by $j$ | $pw_{j,k}$ | $= pref_{j,k} \cdot t_k^C / \sum_k (pref_{j,k} \cdot t_k^C)$ | (B27) |
| Assimilated X={C,N,P,Si} ingestion of | $t_k^X\_z_j^X$ | $= z_j^C \cdot \epsilon_j^X \cdot I_{j,k}^X$ | (B28) |
| Total unass. X={C,N,P,Si} ing. by $z_j$ | $U_j^X$ | $= z_j^C \cdot \sum_k \cdot (1 - \epsilon_j^X) \cdot I_{j,k}^X$ | (B29) |

**Table B5.** Parameters of the zooplankton module. Where necessary, multiple values were provided for micro- and meso-zooplankton. Sources: H97: based on the range provided by Hansen et al. (1997); S97: based on Straile (1997); K17: Kerimoglu et al. (2017b); L12: Lorkowski et al. (2012); RF: Redfield ratio; A: assumed; C: calibrated.

| Symbol | Description | Value$_j$ | Unit | Source |
|---|---|---|---|---|
| $I_{max,j}$ | Maximum ingestion rate | 1.8, 1.5 | - | H97 |
| $K_j^C$ | Half saturation constant | 15.0, 20.0 | - | H97 |
| $Q_j^N$ | Constant N:C ratio of $z_j$ | 0.15 | molN molC$^{-1}$ | RF |
| $Q_j^P$ | Constant P:C ratio of $z_j$ | 0.0094 | molP molC$^{-1}$ | RF |
| $\epsilon_j^C$ | C Assimilation efficiency | 0.5, 0.4 | - | S97 |
| $\epsilon_j^{N,P}$ | N&P Assimilation efficiency | 0.8, 0.8 | - | A |
| $e_j$ | Excretion rate | 0.05 | d$^{-1}$ | A |
| $m1_j$ | Linear mortality rate | 0.02 | d$^{-1}$ | C |
| $m2_j$ | Quadratic mortality rate | 0.01, 0.02 | d$^{-1}$/(mmolC m$^{-3}$) | C |
| $\delta_{S,j}$ | Fraction of mort. & unass. ing. diverted to small det. | 0.85, 0.7 | - | L12 |
| $\delta_{dom}$ | Fraction of DOM in unassimilated ingestion | 0.8 | - | A |

**Table B6.** Assumed grazing preferences $pref_{j,k}$ of predator $j$ (rows) for target $t_k$ (columns).

| | $det_S$ | $p_{flag}$ | $p_{diat}$ | $z_{mic}$ |
|---|---|---|---|---|
| z$_{mic}$ | 0.4 | 0.5 | 0.1 | - |
| z$_{mes}$ | - | 0.3 | 0.1 | 0.6 |

## B2   Abiotic component

### B2.1   Organic material and nutrients

The abiotic components, describe the geochemical transformation between various organic and inorganic pools (DIM, DOM, small and large detritus classes, O$_2$ and the particulate organic matter in the benthos (B-POM), see Fig. 2). Model structure

used here is simplified from ECOHAM (Lorkowski et al., 2012), by excluding carbonate dynamics entirely and simplifying the description of DOM remineralization. The latter is described here as a first order kinetic reaction (eq. B43), instead of a more detailed description of scavenging of DOM by bacterial biomass in the original model. Coupling of the abiotic component with the planktonic components are mediated through the uptake of DIM by phytoplankton (B3,B4), and the recycling of the dead and surplus material. The unassimilated fraction of the ingestion by zooplankton are distributed into the DOM (B35) and the two detritus pools as in Lorkowski et al. (2012). For X=C,N,P, mortality of plankton (B18,B25) are distributed into the small and large detritus classes (B36,B37). For Si, there are no DOM or $det_S^{Si}$ pools (Fig.2), therefore all diatom mortality and Si bound to the ingested diatoms are diverted to the $det_L^{Si}$ (B38).

Conversion of areal units (mmolX/m$^2$/d) of the surface and bottom flux terms (B52-B56) to volumetric units (mmolX/m$^3$/d) required for the pelagic variables is handled by the FABM coupler through division by the surface and bottom layer thicknesses ($\Delta z(s), \Delta z(b)$) internally, which are specified here but not in the model codes.

**Table B7.** Source-sink terms of the dynamic variables (all in mmol/m$^3$/d, except for the benthic variables (B39-B40) in mmol/m$^2$/d) of the abiotic module. Description of processes or functional relationships and parameters are provided in Tables B9 and B8, respectively.

| | | |
|---|---|---|
| DINO3 | $s(DINO3) = DINH4\_DINO3 - \sum_i DINO3\_p_i^N - DINO3\_BPOM/\Delta z(b) - DINO3\_N2$ | (B30) |
| DINH4 | $s(DINH4) = BPON\_DINH4/\Delta z(b) + DON\_DINH4 + sum_j z_j^N\_DIN - \sum_i DINH4\_p_i^N - DINH4\_DINO3$ | (B31) |
| DIP | $s(DIP) \quad = BPOP\_DIP/\Delta z(b) + DOP\_DIP + sum_j z_j^P\_DIP - \sum_i DIP\_p_i^P$ | (B32) |
| DISi | $s(DISi) \quad = BPOSi\_DISi/\Delta z(b) + det_L^{Si}\_DISi - \sum_i DISi\_p_i^{Si})$ | (B33) |
| O2 | $s(O2) \quad = air\_O2/\Delta z(s) + \sum_i p_i\_O2 - \sum_j O2\_z_j - O2\_DOM - O2\_DINH4 - O2\_BPOM$ | (B34) |
| Diss. org. X={C,N,P} | $s(DOX) \quad = \sum_i p_i^X\_DOX + \sum_j (\delta_{dom} \cdot U_j^X) + \sum_{C=S,L} det_C^X\_DOX - DOX\_DIX$ | (B35) |
| Small det. X={C,N,P} | $s(det_S^X) \quad = \sum_i (\delta_{S,i} \cdot M_i^X) + \sum_j (\delta_{S,j} \cdot ((1-\delta_{dom}) \cdot U_j^X + M_j^X) - det_S^X\_DOX - det_S^X\_BPOX/\Delta z(b)$ | (B36) |
| Large det. X={C,N,P} | $s(det_L^X) \quad = \sum_i ((1-\delta_{S,i}) \cdot M_i^X) + \sum_j ((1-\delta_{S,j}) \cdot ((1-\delta_{dom}) \cdot U_j^X + M_j^X) - det_L^X\_DOX - det_L^X\_BPOX/\Delta z(b)$ | (B37) |
| Large det. Si | $s(det_L^{Si}) \quad = \sum_i (M_i^{Si}) + \sum_j U_j^{Si} - det_L^{Si}\_DISi - det_L^{Si}\_BPOSi/\Delta z(b)$ | (B38) |
| Benthic-POX={C,P,Si} | $s(BPOX) \quad = \sum_{c=S,L} det_c^X\_BPOX - BPOX\_DIX$ | (B39) |
| Benthic-PON | $s(BPON) \quad = \sum_{c=S,L} det_c^N\_BPON - BPON\_DINH4 - BPON\_N2$ | (B40) |

## B2.2 Light

In GETM, light intensity at a given depth, $I(z)$, is described by:

$$I(z) = I_0 \cdot a \cdot \exp(-\frac{z}{\eta_1}) + I_0 \cdot (1-a) \cdot \exp(-\frac{z}{\eta_2} - \int_z^0 \sum_n K_n(z')dz') \tag{B62}$$

where, $I_0$ is the light at the water surface, $a, \eta_1$ and $\eta2$ describe the attenuation of the red and blue-green spectra, and $K_n$ describe various constituents in the water, i.e., phytoplankton, detritus, DOC and SPM. For the former three, concentrations of

**Table B8.** Parameters of the abiotic module. Sources: L12: Lorkowski et al. (2012); E5C: ECOHAM5 source code; S96: Seitzinger and Giblin (1996); A: assumed; C: calibrated.

| Symbol | Description | Value | Unit | Source |
|---|---|---|---|---|
| $\lambda^X$ | Rem. rate of DOM | 0.05 | $d^{-1}$ | C |
| $r_S^{N,P}$ | Decay rate of N&P in small det. | 0.12 | $d^{-1}$ | L12 |
| $r_S^C$ | Decay rate of C in small det. | $r_S^{N,P} \cdot 0.85$ | $d^{-1}$ | L12 |
| $r_L^{N,P}$ | Decay rate of N&P in large det. | 0.1 | $d^{-1}$ | L12 |
| $r_L^C$ | Decay rate of C in large det. | $r_S^{N,P} \cdot 0.85$ | $d^{-1}$ | L12 |
| $r_L^S i$ | Decay rate of Si in large det. | $r_S^{N,P} \cdot 0.085$ | $d^{-1}$ | E5C |
| $r_{nit}$ | Nitrification rate | 0.05 | $d^{-1}$ | C |
| $QN_b$ | Bacterial N:C ratio | 0.25 | molN/molC | L12 |
| $w_{detS}$ | Sinking rate of small det. | 2.0 | $m\ d^{-1}$ | C |
| $w_{detL}$ | Sinking rate of large det. | 10.0 | $m\ d^{-1}$ | L12 |
| $br^C$ | Benthic rem. rate of C | 0.028 | $d^{-1}$ | L12 |
| $br^{N,P}$ | Benthic rem. rate N&P | 0.0333 | $d^{-1}$ | L12 |
| $br^S i$ | Benthic rem. rate of Si | 0.0130 | $d^{-1}$ | L12 |
| $\rho_{Seitz}$ | Denit./O$_2$ cons. prop. constant | 0.116 | molN/molO$_2$ | S96 |
| $\omega_{detS}$ | Sed. rate of small det. | $0.25 \cdot w_{detS}$ | $m\ d^{-1}$ | C |
| $\omega_{detL}$ | Sed. rate of large det. | $0.5 \cdot w_{detL}$ | $m\ d^{-1}$ | C |

which are explicitly modelled, $K_n = k_n \cdot C_n$, where $C_n$ is the concentration of constituent $n$, and $k_n$ is the specific attenuation coefficients, which are set to $k_{p_i^C} = 0.015$, $k_{det^C} = 0.01$ and $k_{DOC} = 0.002 \, m^2 \, mmolC^1$ (Oubelkheir et al., 2005; Stedmon et al., 2001). For describing the contribution of SPM, $K_{SPM}$, which is not explicitly modeled here, we use an analytical function of the form:

$$K_{SPM} = K'_{SPM} \cdot f_{SPM}(z) \cdot f_{SPM}(t) \tag{B63}$$

where, the $K'_{SPM}$ is the maximum potential attenuation, $f_{SPM}(z_{max})$ ($z_{max}$: bottom depth) is a sigmoidal function of depth to account for the cross-shore variations and $f_{SPM}(t)$ ($t$: day of year) is a sinusoidal function to account for the cyclic seasonal variations driven by the riverine discharges at the coastal region and thermal stratification offshore:

$$f_{SPM}(z_{max}) = fz_{minfr} + (1.0 - fz_{minfr}) * (1.0 - 1.0/(1.0 + \exp(z*_{max} - z_{max} * 0.5))) \tag{B64}$$

$$f_{SPM}(t) = F * (A * sin(2.0 \cdot t \cdot \pi/365.0 + 2.0 \cdot L \cdot \pi/365.0) + B) \tag{B65}$$

Based on an analysis (see Kerimoglu, 2014) of the temporally and spatially variable SPM data collected by a Scanfish device (see Maerz et al. (2016) for a description of the data set), and the model performance, we fitted $K'_{SPM} = 1.5$, $fz_{minfr} = 0.3$, $z*_{max} = 7.5$ and $F = 0.05$, $A = 6.0$, $B = 12.0$, $L = 85.0$ for $f_{SPM}(t)$. Finally, for the parameterization of $a, \eta_1$ and $\eta_2$, we specify the Jerlov Type-1 option in GETM , which corresponds to clear ocean waters (Paulson and Simpson, 1977), given that we explicitly take the attenuation by organic and SPM constituents into account.

**Table B9.** Process equations and functional relationships used in the abiotic module.

| | | | |
|---|---|---|---|
| $O_2$ switch | $SW_{O2}$ | if $O2 > 0.0 = 1.0, \text{else} = 0.0$ | (B41) |
| $NO_3$ switch | $SW_{NO3}$ | if $DINO3 > 0.0 = 1.0, \text{else} = 0.0$ | (B42) |
| Remineralization of DOX | $DOX\_DIX$ | $= f_{T,abio} \cdot \lambda^X \cdot DOX$ | (B43) |
| Decay of X={C,N,P} in $\det_S$ | $det_S^X\_DOX$ | $= f_{T,abio} \cdot r_S^X \cdot det_S^X$ | (B44) |
| Decay of of X={C,N,P,Si} in $\det_L$ | $det_L^X\_DOX$ | $= f_{T,abio} \cdot r_L^X \cdot det_L^X$ | (B45) |
| Nitrification of pelagic $NH_4$ | $DINH4\_DINO3$ | $= SW_{O2} \cdot f_{T,abio} \cdot DINH4 \cdot r_{nit}$ | (B46) |
| Denitrification in water | $DINO3\_N2$ | $= 0.5 \cdot (1 - SW_{O2}) \cdot SW_{NO3} \cdot DOC\_DIC \cdot QN_b$ | (B47) |
| $O_2$ production by $p_i$ | $p_i\_O2$ | $= DIC\_p_i^C \cdot 1.0[\text{molO2/molC}]$ | (B48) |
| $O_2$ consumption by $z_j$ | $O2\_z_j$ | $= z_j^C\_DIC \cdot 1.0[\text{molO2/molC}]$ | (B49) |
| $O_2$ consumption by remin. | $O2\_DOM$ | $= SW_{O2} \cdot DOC\_DIC + (1 - SW_{O2}) \cdot (1 - SW_{NO3}) \cdot DOC\_DIC$ | (B50) |
| $O_2$ consumption by nitrif. | $O2\_DINH4$ | $= DINH4\_DINO3 \cdot 2.0[\text{molO2/molN}]$ | (B51) |
| $O_2$ flux from air | $air\_O2$ | $= k(O2_0 - O2)$, $k$ from Wanninkhof (1992), $O2_0$ from UNESCO (1986) | (B52) |
| Sedimentation of $\det_S^X$ | $det_S^X\_BPOX$ | $= \omega_{detS} \cdot det_S^X$ | (B53) |
| Sedimentation of $\det_L^X$ | $det_L^X\_BPOX$ | $= \omega_{detL} \cdot det_L^X$ | (B54) |
| Benthic X={C,P,Si} remin. | $BPOX\_DIX$ | $= br^X \cdot BPOX$ | (B55) |
| Benthic $O_2$ consumption | $O2\_BPOM$ | $= SW_{O2(b)} \cdot BPOC\_DIC + (1 - SW_{O2(b)}) \cdot (1 - SW_{NO3(b)}) \cdot BPOC\_DIC$ | (B56) |
| Potential benthic denit. | $BPON\_N2'$ | $= \rho_{Seitz} \cdot O2\_BPOM$ | (B57) |
| Benthic denitrification | $BPON\_N2$ | $= BPON\_N2' - \max(0.0, BPON\_N2' - BPON\_DINH4')$ | (B58) |
| Potential benthic N remin. | $BPON\_DINH4'$ | $= br^N \cdot BPON$ | (B59) |
| Benthic N remineralization | $BPON\_DINH4$ | $= \max(0.0, BPON\_DINH4' - BPON\_N2')$ | (B60) |
| Benthic $NO_3$ reduction | $DINO3\_BPOM$ | $= 0.5 \cdot SW_{O2} \cdot SW_{NO3} \cdot BPON\_DINH4$ | (B61) |

*Author contributions.* OK designed the study with contributions from YV, developed the biogeochemical model, performed the model skill assessment and conducted the model-based analyses with contributions by FC and prepared the first draft of the manuscript. JvB provided the monthly average nutrient concentrations, and YV assisted the compilation of the monitoring data. RH and KK assisted the improvement of the hydrodynamical model with horizontal diffusion. All co-authors contributed to the discussion of the results and revisions of the manuscript.

*Competing interests.* The authors declare that no competing interests are present.

*Acknowledgements.* OK was supported by the German Environment Agency, UBA [3718252110] and German Science Foundation, DFG [KE 1970/1-1 and KE 1970/2-1]. OK and FC was supported by the German Federal Ministry of Research and Education, BMBF [03F0740B].

KK was financed by the Collaborative Research Centre of the DFG, TRR 181 [274762653]. Data at Helgoland were provided by Alfred Wegener Institute (AWI), and at Norderelbe, Suederpiep and Westerhever were provided by the Landesamt für Landwirtschaft, Umwelt und ländliche Räume des Landes Schleswig Holstein. Noordwijk and Terschelling data were provided by the Rijkswaterstaat. Simulations were performed on the 'Mistral' supercomputer of the German Climate Computing Center (DKRZ). We thank Johannes Pätsch (Uni. Hamburg) for providing the boundary conditions and ECOHAM code that helped developing the abiotic component of the model, Sonja van Leeuwen

(NIOZ) for providing the riverine data, Wilhelm Petersen, Gisbert Breitbach and KOI group of the HZG for the acquisition and maintenance of the FerryBox and COSYNA data and Ivan Kuznetsov (then at HZG) for the cooperation on model validation. OK is grateful to the developers of the open source platforms and free software that made this work possible, foremost, GETM, GOTM, FABM, NetCDF, NCO, Python, Kubuntu OS and UNIX tools. We thank H.-J. Lenhart, J. Pätsch and two anonymous referees for their valuable comments.

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
