# Peer review of "Interactive impacts of meteorological and hydrological conditions on the physical and biogeochemical structure of a coastal system"

_Biogeosciences, 2020_

## Referee Comment (RC1) · Anonymous Referee #1 · 11 Mar 2020

**Review** of the manuscript *Interactive impacts of meteorological and hydrological conditions on the physical and biogeochemical structure of a coastal system*, by Kerimoglu et al, submitted to **Biogeosciences**

**Manuscript overview**

The manuscript provides a model study into a particular flooding event in Northern Germany in order to determine the driving forces with regard to the marine response (German Bight area) to the event. To this end a slightly altered model is presented and applied under 2012 and 2013 conditions, plus 3 scenarios for 2013 to test the different, expected driving forces (meteorology, riverine input and a particular, 2 months long wind regime). The authors first show the anomalous forcing events, followed by plenty of model validation results and finally the model study into the expected drivers, for which they analyse the abiotic and biotic response of the system. They then conclude that the marine response to the flooding event was determined by both the enhanced riverine input (fresh water, nutrients, inner German Bight) and the anomalous meteorology of 2013 (outer German Bight) interacting with each other to alter the estuarine circulation patterns within the area.

The appendix contains detailed information about changes to the hydrodynamic and biogeochemical model, including applied equations and parameter values. It also contains some more validation results to justify some of the changes made to the model.

**Review overview**

In all, I'm quite charmed by the paper's objective and presented model study, with the specific aim (rather than a hypothesis) to determine which factors led to the marine reaction to a particular land-based event. The approach is valid and very interesting from a physical point of view. But I miss a good spatial validation of the model (surely the data in Figure 5 can provide that) which would more clearly quantify the problems in simulating the near shore environment.  Also, I'm not quite sure why a new model is presented which doesn't include bacteria in a study that aims to understand dissolved oxygen issues in the area. Why not use ECOHAM for that? Or better, a model with benthos included? The authors build on earlier work, and explicitly state that they use a simplified version of ECOHAM from which carbonate and bacterial dynamics have been eliminated (line 484, the geochemical model). They base their new biogeochemical model (does it have a name?) on a previous model by the lead author that included mixothrophs, but these are not in here. So we have a biogeochemical model with just 2 phytoplankton species, 2 zooplankton species, fixed nutrient ratios inside zooplankton (the regulated uptake described in B1), no bacteria and no benthos. Isn't that just a stripped version of ECOHAM? Why not use that model? And if an important feature has been added (e.g. variable N,P ratio within phytoplankton), why not add it to ECOHAM? I would also argue there are more complex models out there better suited for a dynamic, shallow area like the German Bight, particularly for a study involving nutrient concentrations and bottom oxygen conditions. Given the lack of validation with Chla observations (the only station in the area of interest shows a normalized model bias of 1.12) and benthic nutrient concentrations my confidence in the biogeochemical model results is not large. Although the authors are in parts clear about the model limitations, they should add text on 1. Their choice of biogeochemical model, 2. What makes it better suited here than ECOHAM, 3. More Chla validation and 4. The role the sediments play in nutrient dynamics in shallow areas. Or, as an alternative, the authors could limit their analysis to the physical

part, which is quite strong in the manuscript and would allow for a better focus of the text: there is enough to analyse there as shown by the authors, and the conclusions would not change.

**Recommendation**

Major revision

**Detailed Comments**

L 56-57: One cannot expect that the marine transport of riverine inputs is purely dependent on the inter-annual variability in the river discharges. In any marine area the meteorological conditions (mainly wind and temperature) will play a large part in the transport, as will alongshore currents. Then there are influences like mixing by ships, the presence of off-shore wind farms, and further-afield influences like the Rhine discharge. So I thought this sentence a little odd.

Fig. 2: The diagram is clear until one gets to the appendices, where it is stated that phytoplankton exudates DOM (L463), that zooplankton excrete into the DIM pool (L466) and the unassimilated fraction ingested by zooplankton becomes DOM (L486). None of this is visible in the model diagram, as all functional groups just exude large detritus … ?

L 101: The authors state here that the underwater light conditions are determined by detritus, DOM and a background value representing SPM. But in section B2.2 they state that phytoplankton is also included in the light calculation. Please make this consistent.

L 116: Please provide the website for the atmospheric deposition fields.

L 117: Please state which rivers were included within the Wadden Sea area. Just major ones (Elbe, Weser, Ems, …) or also local Dutch and German rivers like the Accumersiel, Bensersiel, Wangersiel, Miele, etc.? I know from experience that these rivers are also part of the mentioned database, which I think is called the OSPAR ICG-EMO riverine database. So I would assume they were used, but this needs to be stated clearly.

L 124: "a 3600 s time window", why not say 1 hour time window? In the caption of Figure 4 the authors mention an hourly resolution, not a 3600 s one.

L 134: Again, a website for the ICES data should be provided.

L 139: This section is called Results, but quite a large part of it is model validation results. I would like to see this separate from the forcings analysis (section 3.3 onwards), and would therefore call this section "Model validation" and rename section 3.3 to be section 4 "Results".

L 144: Naturally the nutrient loads follow the flow peak, but what about concentrations? If we assume heavy rainfall caused more run-off then nitrogen concentrations may stay the same, but phosphorous concentrations (usually from sewage treatment works) may be diluted. So please provide some measure of the changes in concentrations for these rivers.

Fig. 3: The Ems does not show the flood peak found in the Weser and the Elbe, suggesting it was a local event. Nevertheless I would like to see results for the Rhine/Meuse system, which will influence the area of interest here under normal conditions.

L. 146-150: Please provide some information on whether 2012 was in any way an average year or not.

Fig. 4: It seems that 2013 is characterized by mainly eastern winds all the way up to June. So why were only the June-August winds selected for a scenario? Because they do not seem easterly much in that

period. The winter and spring easterlies are now part of the M12 scenario, together with the different temperature record etc.

Fig.5 : Please make this a colour graphs, the gray scales are very hard to distinguish from one another. And why is count on the colour bar at all? I assume this is the number of observations in a given point throughout the year? But why not use three different colours for the three years instead?

Fig. 5: And as said before, I would really like to see a spatial validation graph, which would provide more detail on the nearshore errors in the model. I realise there are quite a large number of figures already in this manuscript, but would suggest some could be put in the appendix, e.g. Figure 6 and Figure 8 (which shows 3 stations which are in the model domain but not in the area of interest, and which therefore do not provide much context for the described work).

L162: Why use Kelvin here when Fig. 4 uses Celsius?

L175: The authors state that the plume was realistically reproduced as the sharp increase in NO3 al Helgoland was captured. But this is not very clear from Fig. 8, rather that 2 observed peaks in DIN are not reproduced by the model and one peak is slightly reproduced. So I'm not convinced that the plume is simulated realistically, just from this figure.

L177: why do the authors have such a high Si value on the western boundary? Is this an artefact of the simulation that generated the boundary conditions?

L181: The model fails to get the spring bloom timing right. I would say: use a different model or just focus on the physics. The Chla comparison for Helgoland is quite bad and this is the only station presented here for validation of Chla in the area of interest. Does ICES have more Chla data in the specific area?

Fig 10: This figure, and also figures 11 and 12 are too small for readers to easily read. I would suggest that the graph itself is made larger in the manuscript but also that the colour bar is changes to one large one on each side (one for S, one for T), so the graph becomes more accessible. These graphs are the essential results presented in the manuscript, so please do them justice.

L223: It is not clear to me why increased stability should have a direct effect on the underwater light penetration, particularly as SPM dynamics are just a background value. Are the authors referring here to limited nutrient exchange and thus less bioshading? They do for the OGB, but in the CGB the flood causes increased stratification and brings in nutrients, resulting in more primary production,. On L246 it is simply stated that increased stratification enhanced the underwater light regime within the CGB. Please explain and provide a reference. Are you referring to increased remineralization within the euphotic zone? I would also like to see some evidence of the underwater light response in the simulations.

L248: Please introduce figure 13 first and explain the DO abbreviations before going into the analysis.

Fig. 6: Can the authors speculate why their biogeochemical model is unable to quantitatively reproduce the observed oxygen minimum? What processes do they think the model misses?

Fig. 15: These again are too small and I cannot see the arrows at all in the difference figures.

L282: Yes, they do but this is rather an open door. Any reader would have expected that from the start, and would have been surprised if this was not the case.

Sec4: Please discuss the lack of bacterial dynamics in the discussion, and the effect this can have on the simulated results.

L312: I would say the model was able to reproduce the *physical* characteristic features of the system quite well.

L316: "The skill of the model … is notable", quite a nice notation as it is meaningless. Notable means it can be noted, it says nothing about it being good or bad.

L320-333: I'm not sure why this is include here, this is not of interest for the general reader I would think. Therefore I would put this in an appendix at most.

L349: I fail to see the prolonged stratification in figure 11. As these are all July averages I don't see a time indication in this figure at all.

L385: I object to the use of the word "satisfactorily" when it comes to the reproduction of the biogeochemical features of the German Bight ecosystem.

Table B5,B6,B8: If parameter values are provided then references on what these are based on should be included as well. Assuming these values have not been published before.

**Language**

In general I found the manuscript very readable, yet the English used was not always correct or as expected. I found several mistakes regarding single/plural (e.g. L 140, "the discharge rates … peak**s**", L156 "Comparison … **are** shown", L299 "potential sources of error need**s** to be addressed"), omissions of articles (e.g. L 156 *A* "Comparison of", L187 "Despite a tendency to overshoot, *the* range of", L205 *The* "Effect of exchanging", L211 "further to *the* North"), additions of articles in unnecessary places (e.g. L 143 "over **the** central Europe", LL232 "river forcing of **the** 2012 is used", L232 "the plume of **the** DIP") and omission of connecting words (e.g. L157 "are located at shallow sites, *and* therefore provide", L373 "the presence *of* regional differences"). I suggest the authors check their English thoroughly before the next submission. But I love the double negative found on L439: "leading to near-complete elimination of negative values of the total mixing being removed". So the removal has been eliminated?

---

## Referee Comment (RC2) · Anonymous Referee #2 · 8 Apr 2020

This study introduces a new biogeochemical model, consisting of modified versions of previously published models. Coupled with a hydrodynamic model with an improved mixing scheme, the system is validated in the German Bight region, and used to assess the impact of meteorology and river forcing on a specific flood event in 2013. The conclusion is that an interplay of the two resulted in anomalous conditions, as previously noted in observations.

The paper acts as both presentation and validation of a new modelling system, and an investigation into a specific event. While it could potentially work as separate papers, the paper is well enough written and laid out, and the validation both sufficiently com-

prehensive and targeted, that it works well and is an enjoyable and interesting read. I recommend publication in Biogeosciences subject to a few minor comments detailed below.

The biogeochemical model appears to be a work in progress towards a different mixotroph-based model, rather than a model likely to be widely used in its present form, if I've got the correct impression? This is fine, given its structure seems sensible and plenty of validation is presented, but it would be worth adding some discussion about what sets it apart from other similar models, particularly ECOHAM, and what future developments are intended.

Given that it's central to the study, in Section 2.1 and/or 2.2 it would be worth explicitly detailing which variables are used in the atmospheric and riverine forcing, and how they're applied to the model (e.g. bulk formulae? are rivers applied just at the surface or over the full depth?).

"using the 'spatial.cKDTree' package from the Scipy library of Python 3.5." – add the Scipy version number for completeness.

Figure 3 shows the Ems, and that this doesn't have anomalous discharge in 2013. This isn't mentioned or discussed in Section 3.1, and should be. Also, the "dashed blue lines" appear solid.

"Simulated temperature and salinities ... (Fig. 5) ... exhibit no signs of systematic deviations or biases." The calculated $B^*$ values are near-zero, but by eye it looks like there's a cold bias, particularly at colder temperatures, and that salinity is usually too high. Is this just a trick of the eye, or are the simulated and observed distributions different? Please also state what $B^*$, rho and n are in the caption of Fig. 5, as per Fig. 6.

"(Fig.8) ... The ability of the model to capture the sharp increase in DIN during June/July 2013 at the Helgoland station suggests that the spreading of the plume

of the Elbe-Weser rivers following the flood event was realistically reproduced." The model completely misses the peak earlier in the year, and also in early 2014. Can you be confident therefore that this result was obtained for the right reasons?

In Fig. 7, plotting the average in white is confusing – I initially thought there were separate yellow/blue lines either side of it, and the white was blank space. Plotting it in dark yellow/blue might be better. Also, make clear in the caption that the line indicates the average and the shading the standard deviation (I assume this is the case?).

In Fig. 14 it would be best to avoid plotting red and green together, as this renders it inaccessible to those who are red-green colour blind. (Disclaimer: I'm not colour blind myself, so can't say for sure.)

―――――――――――――――――――

---

## Author Comment (AC1) · 29 Apr 2020

**Response to Anonymous Referee #1**

Below, referee comments (starting with 'Comment'), and our specific responses (starting with 'Response' and/or [envisioned] 'Change') are provided in black and blue fonts, respectively.

**Manuscript overview**
The manuscript provides a model study into a particular flooding event in Northern Germany in order to determine the driving forces with regard to the marine response (German Bight area) to the event. To this end a slightly altered model is presented and applied under 2012 and 2013 conditions, plus 3 scenarios for 2013 to test the different, expected driving forces (meteorology, riverine input and a particular, 2 months long wind regime). The authors first show the anomalous forcing events, followed by plenty of model validation results and finally the model study into the expected drivers, for which they analyse the abiotic and biotic response of the system. They then conclude that the marine response to the flooding event was determined by both the enhanced riverine input (fresh water, nutrients, inner German Bight) and the anomalous meteorology of 2013 (outer German Bight) interacting with each other to alter the estuarine circulation patterns within the area.
The appendix contains detailed information about changes to the hydrodynamic and biogeochemical model, including applied equations and parameter values. It also contains some more validation results to justify some of the changes made to the model.

**Review overview**
In all, I'm quite charmed by the paper's objective and presented model study, with the specific aim (rather than a hypothesis) to determine which factors led to the marine reaction to a particular land-based event. The approach is valid and very interesting from a physical point of view.
Response: We thank the referee for the thorough review and positive remarks.

But I miss a good spatial validation of the model (surely the data in Figure 5 can provide that) which would more clearly quantify the problems in simulating the near shore environment.
Response: The data used for Figure 5 does not provide a homogeneous or balanced spatio-temporal representation within the study area, therefore it does not allow a reliable spatial validation suggested by the referee (see our response to the detailed comment below). However, we are convinced that the provided comparisons of modeled estimates against data from continuous Ferrybox measurements (Fig. 7 and A1), 3 stations for physical variables (Fig. 6), and 7 stations for biological variables (Fig. 8-9, 3 of these in Fig. 9 being near-shore stations in the study area) provide sufficient evidence that the model satisfactorily captures the spatio-temporal variability in the study area relevant for the purposes of the study. Therefore we believe that the presented model performance assessment is sufficient for the purposes of the current study and a more detailed examination of the nearshore variabilities is beyond the scope of this manuscript (see also the responses below to the respective detailed comments).

Also, I'm not quite sure why a new model is presented which doesn't include bacteria in a study that aims to understand dissolved oxygen issues in the area. Why not use ECOHAM for that? Or better, a model with benthos included? The authors build on earlier work, and explicitly state that they use a simplified version of ECOHAM from which carbonate and bacterial dynamics have been eliminated (line 484, the geochemical model). They base

their new biogeochemical model (does it have a name?) on a previous model by the lead author that included mixothrophs, but these are not in here. So we have a biogeochemical model with just 2 phytoplankton species, 2 zooplankton species, fixed nutrient ratios inside zooplankton (the regulated uptake described in B1), no bacteria and no benthos. Isn't that just a stripped version of ECOHAM? Why not use that model? And if an important feature has been added (e.g. variable N,P ratio within phytoplankton), why not add it to ECOHAM?

Response: Some of these questions are suggestive of a number of misunderstandings, possibly led by ambiguities in the model description, which we hope to clarify in the following:

1) It is not entirely clear to us, what is meant by 'why not use ECOHAM for that'. If it means directly applying a readily available ECOHAM setup, this was not an option: to the best of our knowledge, the only available HAMSOM-ECOHAM setup, performance of which have been sufficiently documented (e.g., Große, 2017), is simply too coarse (20 km horizontal resolution, and 7 z-levels within the deepest part of the study area) for being able to capture the meso-scale features of the system, in particular, the haline stratification caused by the Elbe river (Pätsch et al., 2017).

2) If what the referee means is to couple ECOHAM with our GETM setup, which was previously identified to successfully reproduce the hydrodynamics of the study system (Kerimoglu et al., 2017a, Nasermoaddeli et al., 2018), this was not an option either: there is no ECOHAM code that can be readily coupled with GETM.

3) In this study, for the hydrodynamics, we used the GETM setup mentioned above. For the biogeochemistry, we used a model developed based on the earlier work of the first author (Kerimoglu et al., 2017b). In this model, description of the non-planktonic processes (i.e, production and destruction of detritus, oxygen consumption processes, benthic remineralization) were adjusted to the study system by adopting almost the same structure and descriptions provided by ECOHAM. It should be noted that, given that the present model accounts for the variable stoichiometry and chlorophylll content of phytoplankton, adopting the descriptions of plankton growth and interactions from ECOHAM as well would mean a backwards step, technically.

4) In the original ECOHAM model, 'bacterial' oxygen consumption occurs in proportion to the DOM breakdown. In our model, although the bacterial biomass is not explicitly considered, the oxygen consumption in proportion to DOM breakdown is represented. The only difference between the two approaches is that in ECOHAM bacterial abundance is potentially limiting for the DOM breakdown rate, while in ours, it is not (see the detailed response below).

5) It was again not entirely clear to us what is exactly meant by 'a model with benthos' by the referee. To clarify, our model does comprise a benthic module that describes the aerobic and anaerobic early diagenesis in the sediment, exactly as described by ECOHAM. We acknowledge, however that the descriptions of benthic processes provided by this model are simplistic, and potentially responsible for, e.g., inaccuracies in oxygen consumption rates (see below).

6) Although the description of the non-planktonic processes are similar, differences in the descriptions of plankton growth and interactions between our model and ECOHAM are significant. Therefore, referring to our model as 'a stripped version of ECOHAM' would be misleading.

I would also argue there are more complex models out there better suited for a dynamic, shallow area like the German Bight, particularly for a study involving nutrient concentrations and bottom oxygen conditions.

Response: There are certainly more complex models, but considering the purposes of our study, it is not clear in which specific sense would such a model be better suited. It should be noted that, with regard to benthic/pelagic coupling, models of similar complexity have

been used until recently, for studying the nutrient concentrations and bottom oxygen conditions in the North Sea (e.g., Große et al., 2017, using ECOHAM), as well as other similarly dynamic coastal shelf systems such as the Louisiana Shelf (e.g., Fennel and Laurent, 2018) or even shallower systems such as the Chesapeake Bay (Irby et al. 2018).

Given the lack of validation with Chla observations (the only station in the area of interest shows a normalized model bias of 1.12)
Response: the comparison of estimated chlorophyll concentrations with the data from four stations does build confidence in the simulated chlorophyll in the study area: although three of these stations are outside the exact study area, they are still close enough to be representative, as they are characterized by a similar abiotic environment that is typically found in the study area. Although a normalized bias of 1.12 for chlorophyll is obviously not very good (which we openly highlighted and discussed in the manuscript), it is not alarming, considering that chlorophyll is governed by exponential growth dynamics, and therefore commonly shown (when shown at all) in logarithmic scale in model-data comparison plots.

and benthic nutrient concentrations
Response: necessity for the presentation of a validation of benthic nutrient concentrations of fluxes (which is very rarely done in studies similar to ours) is not clear.

my confidence in the biogeochemical model results is not large.
Response: as a clarification, we do not claim confidence in the predictions of our model in an absolute sense, and providing such precise predictions is not our purpose in this study either. However we are confident that the model is useful in gaining insight into the overall response of the ecosystem to changes in hydro-meteorological conditions, which is the purpose of this study (see also our response to point 3 below).

Although the authors are in parts clear about the model limitations, they should add text on 1. Their choice of biogeochemical model, 2. What makes it better suited here than ECOHAM,
Response: Although all technical details extensively listed above are not likely to be relevant for the audience, some clarification of the model design and a discussion of potential future development will serve to improve the general model description.
Change: we will extend the model description section to clarify the model structure, and include a discussion on the similarities with and differences from other models, as well as potential future development and applications. We will also store the model code in a public repository and provide it in the 'Code and data availability' section, so that anyone interested can inspect and use the code.

3. More Chla validation and
Response: we are convinced that the presented validation is already plenty, targeted, and based on an extraordinarily rich dataset.
Change: in line with our view explained above, we will stress in the Discussion section that our results should be interpreted in terms of system response to hydro-meteorological forcing, and not as predictions in an absolute sense.

4. The role the sediments play in nutrient dynamics in shallow areas.
Response: this suggestion is potentially caused by a misunderstanding that our model does not have at all a benthic module (see above). We acknowledge, however, that some complex benthic dynamics,such as the spatial heterogeneities in sediment fluxes driven by sediment permeability are not captured by our simple model, which is potentially

responsible for inaccuracies, e.g., in oxygen consumption rates.
Change: we will include a discussion of these effects.

Or, as an alternative, the authors could limit their analysis to the physical part, which is quite strong in the manuscript and would allow for a better focus of the text: there is enough to analyse there as shown by the authors, and the conclusions would not change.
Response: Referee's suggestion of limiting our analysis to the physics alone, will invalidate roughly half of our conclusions, and hence substantially reduce the scope and significance of our study. We would therefore prefer to keep the analysis regarding  the biogeochemical processes. Please see below our responses to the specific comments.

**Recommendation**
Major revision

**Detailed Comments**
L 56-57: One cannot expect that the marine transport of riverine inputs is purely dependent on the inter-annual variability in the river discharges. In any marine area the meteorological conditions (mainly wind and temperature) will play a large part in the transport, as will alongshore currents. Then there are influences like mixing by ships, the presence of off-shore wind farms, and further-afield influences like the Rhine discharge. So I thought this sentence a little odd.
Response: We do not think that the sentence referred by the referee ('The extent to which the hydrodynamical structure, and the transport of riverine material within the German Bight depends on the inter-annual variability in riverine discharges is not fully understood.'), implies that 'the marine transport of riverine inputs is purely dependent on the inter-annual variability in the river discharges'. The emphasis here is on the not fully understood '*extent*', i.e., the magnitude and scope of this dependency.

Fig. 2: The diagram is clear until one gets to the appendices, where it is stated that phytoplankton exudates DOM (L463), that zooplankton excrete into the DIM pool (L466) and the unassimilated fraction ingested by zooplankton becomes DOM (L486). None of this is visible in the model diagram, as all functional groups just exude large detritus ... ?
Response: The mentioned links were intentionally neglected in an attempt to make the diagram easier to understand.
Change: the  simplifications will be clarified in the caption of Fig. 2 and the reader will be referred to Tables B1-B9 for an accurate model description.

L 101: The authors state here that the underwater light conditions are determined by detritus, DOM and a background value representing SPM. But in section B2.2 they state that phytoplankton is also included in the light calculation. Please make this consistent.
Change: shading caused by phytoplankton will be mentioned in the sentence.

L 116: Please provide the website for the atmospheric deposition fields.
Response: The website is provided in the 'Code and data availability' section. (L401), along with a number of other data sources. We do not think that duplicate listing of these sources in the main text is necessary.

L 117: Please state which rivers were included within the Wadden Sea area. Just major ones (Elbe, Weser, Ems, ...) or also local Dutch and German rivers like the Accumersiel, Bensersiel, Wangersiel, Miele, etc.? I know from experience that these rivers are also part of the mentioned database, which I think is called the OSPAR ICG-EMO riverine database. So I would assume they were used, but this needs to be stated clearly.

Response: in this study, we only used the discharges from major rivers shown in Fig. 1. Previously, we had observed that inclusion of small rivers did not make an appreciable difference in the present setup.
Change: it will be mentioned here that the dataset is indeed called OSPAR ICG-EMO riverine database, and that we considered only the major rivers as shown in Fig. 1.

L 124: "a 3600 s time window", why not say 1 hour time window? In the caption of Figure 4 the authors mention an hourly resolution, not a 3600 s one.
Response: 3600s is how it is specified from a drop-down list in the web-interface of the cosyna data portal, which we thought could have been relevant.
Change: we will use 'hourly' for the sake of consistency.

L 134: Again, a website for the ICES data should be provided.
Response: The website was already provided in the 'Code and data availability' section.

L 139: This section is called Results, but quite a large part of it is model validation results. I would like to see this separate from the forcings analysis (section 3.3 onwards), and would therefore call this section "Model validation" and rename section 3.3 to be section 4 "Results".
Response: The material we present in 3.2 is not a model validation without context, but it is partially targeted towards assessing the ability of the model to capture the flood event specifically (Fig.7,9, and partially Fig.6). Therefore it is important that this section follows the 'Hydrological and Meteorological Conditions' section, which are also clearly part of the Results. It is not clear, what the benefit of separating section 3.2 from the rest of the results would be. Therefore, we would prefer to keep the structure of the manuscript as it is, which we believe to be well connected and easy to follow.

L 144: Naturally the nutrient loads follow the flow peak, but what about concentrations? If we assume heavy rainfall caused more run-off then nitrogen concentrations may stay the same, but phosphorous concentrations (usually from sewage treatment works) may be diluted. So please provide some measure of the changes in concentrations for these rivers.
Change: we will check and report any significant changes in concentrations, or the lack thereof.

Fig. 3: The Ems does not show the flood peak found in the Weser and the Elbe, suggesting it was a local event. Nevertheless I would like to see results for the Rhine/Meuse system, which will influence the area of interest here under normal conditions.
Response: it was mentioned in the text (L.141-143) that the flood event was caused by an event over central Europe, that affected the basins of Elbe and Weser rivers. However, it is indeed not clear from this explanation, whether other rivers may have been affected or not.
Change: we will analyze and specify which rivers are affected.

L. 146-150: Please provide some information on whether 2012 was in any way an average year or not.
Change: we will consider showing the decadal averages in the Figure, but if this makes the figure overly complicated, provide information in the text.

Fig. 4: It seems that 2013 is characterized by mainly eastern winds all the way up to June. So why were only the June-August winds selected for a scenario? Because they do not seem easterly much in that period. The winter and spring easterlies are now part of the

M12 scenario, together with the different temperature record etc.

Response: the point we aim to make with W12 scenario is that the short term wind forcing is so important for the system that the wind forcing only during summer, regardless of the earlier forcing (including wind direction), can make many patterns (especially stratification) resemble those in 2012 (e.g., L.205-206, L.215-220, L.340). Including a longer time period would erode the strength of the scenario by bringing in additional complexities.

Fig.5 : Please make this a colour graphs, the gray scales are very hard to distinguish from one another. And why is count on the colour bar at all? I assume this is the number of observations in a given point throughout the year? But why not use three different colours for the three years instead?

Response: these plots are two-dimensional histograms, where counts represent the frequency of observation-simulation pairs. Higher count (darker shades) simply indicates higher density of pairs, which does not need an exact perception.

Change: we will clarify in the caption that these are two-dimensional histograms, and that counts represent frequency of observation pairs.

Fig. 5: And as said before, I would really like to see a spatial validation graph, which would provide more detail on the nearshore errors in the model. I realise there are quite a large number of figures already in this manuscript, but would suggest some could be put in the appendix, e.g. Figure 6 and Figure 8 (which shows 3 stations which are in the model domain but not in the area of interest, and which therefore do not provide much context for the described work).

Response: Both Fig. 6 and Fig. 8 are essential for the manuscript. All 3 stations in Fig. 6 are within the study domain and show that the model mostly accurately reproduced the measured temperatures and salinities. Although the three stations in Fig. 8 is not within the area of interest, they are quite close and constrained by an abiotic environment (resource abundances, water depth, meteorological and physical conditions) similar to that in the study area, therefore they help building confidence in the biogeochemical model within the study area. In fact, by demonstrating that the model is able to reproduce the baseline levels of measurements obtained at different stations, these plots serve in gaining insight into the model's skill in reproducing cross-shore gradients, which is what the referee probably wants to see with the 'spatial validation graph'. Finally, Fig. 9, which shows the modelled and measured nutrient concentrations at stations located along the coastline downstream of the mouth of Elbe, serves in evaluating the skill of the model in reproducing the spatial distribution of nutrients following the flood event. We are therefore convinced that the presented analyses provide an extraordinarily good basis for the assessment of the performance of the model and provide evidence for its suitability for the purposes of this study.

L162: Why use Kelvin here when Fig. 4 uses Celsius?

Response: In Fig.4, the context is absolute air temperature, where Celsius is an arguably more convenient scale than Kelvin. In the context of a temperature difference Kelvin is practically identical to Celsius but Celsius may indeed be familiar for the general audience.

Change: we will replace K with Celsius in the text.

L175: The authors state that the plume was realistically reproduced as the sharp increase in NO3 al Helgoland was captured. But this is not very clear from Fig. 8, rather that 2 observed peaks in DIN are not reproduced by the model and one peak is slightly reproduced. So I'm not convinced that the plume is simulated realistically, just from this figure.

Response: for convenience, we show in Fig. R1-1 below an enlarged and annotated

version of the related panel in Fig.8 of our manuscript. As can be more clearly seen here, the distinctive 'sharp increase in DIN during June/July 2013' (as stated in L.175) is indeed realistically captured by the model (please see also the Fig. R2-1 included in our response to Referee #2 regarding a related comment, where we show that the ability of the model to capture the DIN peak after the flood is closely coupled with the ability of the model in capturing the freshwater plume of the flood).
Change: we will reformulate this sentence and spell out our take on this particular result.

[Figure]

*Figure R1-1: DIN Concentrations measured (gray dots) and modelled (black line) at the Helgoland station (modified from Fig. 8 in the manuscript).*

L177: why do the authors have such a high Si value on the western boundary? Is this an artefact of the simulation that generated the boundary conditions?
Response: as explained in L.176-178, overestimated Si values are indeed caused by the fluxes from the western open boundary, which is due to too high concentrations specified as the boundary conditions.
Change: we will specify that this is caused by the available data used to specify the boundary conditions.

L181: The model fails to get the spring bloom timing right. I would say: use a different model or just focus on the physics. The Chla comparison for Helgoland is quite bad and this is the only station presented here for validation of Chla in the area of interest. Does ICES have more Chla data in the specific area?
Response: In our view the immense effort of changing the biogeochemical model is not justified for the scope of the present study. Focusing only on physics would mean removal of roughly a half of the presented material, which we consider to be relevant and useful. Opting for any of these paths would require very substantial reasons for doing so, which we do not see.
1) The model indeed fails to capture the timing of the spring bloom at Helgoland, as was mentioned in the L.181 of the manuscript, however this is not directly relevant to the subject matter of the manuscript.
2) We disagree that the model comparison in Helgoland is 'bad', when put in the right context: we are not aware of any other model that shows better performance at this particular station.
3) Comparison at other stations build confidence in model results, even if they are not directly in the area of interest. As mentioned above, these comparisons show that the higher concentrations at the coastal stations and lower concentrations at the off-shore stations are reproduced, which can be expected to hold within the study area.
4) Our biogeochemical model offers many other useful insights into other variables such as nutrient and oxygen concentrations, which are all essential for the manuscript.

5) We acknowledge that the relatively poorer model performance regarding chlorophyll (relative to the other variables), requires a more careful interpretation of directly relevant model results, such as the primary production estimates.

6) ICES dataset offers chlorophyll measurements, however, as shown in Fig. R1-2 below, the spatio-temporal distribution of the reliable (having consistent metadata) data available within the study area is so heterogeneous, that, a construction of, for instance a 'summer average' map with the data will be heavily influenced by the sampling frequency in time and space. Therefore it is not straightforward to achieve a consistent validation with this data set.

Change: Per point 5 in our response above, we will stress in the discussion that the NPPR estimates, which are directly related with overestimated chlorophyll values, need to be interpreted with care. In particular, we will state that rather than the absolute magnitude, the response of NPPR to the hydro-meteorological conditions should be regarded.

[Figure]

*Figure R1-2: Spatial distribution of reliable chlorophyll measurements in the ICES dataset during July and August for each simulation year. n is the number of unique locations (identified by latitude-longitude pairs rounded to nearest 0.05o).*

Fig 10: This figure, and also figures 11 and 12 are too small for readers to easily read. I would suggest that the graph itself is made larger in the manuscript but also that the colour bar is changes to one large one on each side (one for S, one for T), so the graph becomes more accessible. These graphs are the essential results presented in the manuscript, so please do them justice.

Response: the particular suggestion of the referee can indeed be applied to Fig. 10, but not equally well to Fig. 11 and not at all to Fig. 12, as in the latter, not only two, but four variables need to be shown. We do not think that presenting these figures with different layouts will improve the manuscript. Considering that the latitude and longitudes are presented in larger form in Fig. 1, these labels can be removed to save space for larger colorbars. When these Figures are printed in full page width in the final publication, the text will be presumably easier to read as well: for the discussion paper, they are constrained to 12cm width, as was suggested by the style guidelines.

Change: The latitude and longitude labels will be removed and colorbars will be enlarged in Figures 10-13.

L223: It is not clear to me why increased stability should have a direct effect on the underwater light penetration, particularly as SPM dynamics are just a background value. Are the authors referring here to limited nutrient exchange and thus less bioshading? They do for the OGB, but in the CGB the flood causes increased stratification and brings in nutrients, resulting in more primary production,. On L246 it is simply stated that increased stratification enhanced the underwater light regime within the CGB. Please explain and provide a reference. Are you referring to increased remineralization within the

euphotic zone? I would also like to see some evidence of the underwater light response in the simulations.

Response: we would like to clarify that in this sentence ('intensity of the thermohaline stratification, [and hence], gives insight into the average light conditions primary producers experience in the deeper zones') there was a typo: 'deeper' should be in fact 'surface'. As it may now have become clear after this correction, with this sentence, we were not referring to the changes in the 'underwater light penetration', but simply to the obvious fact that, due to the reduced vertical mixing, phytoplankton growing at the surface layers can stay there longer, enhancing therefore the 'light conditions [they] experience'. Due to the large uncertainties in the underwater light climate, and only the partial coverage of its response to the hydroclimatological factors (e.g., see L.359-361 in Discussion), we would not like to present potentially misleading estimates.

Change: 'deeper zones' in the sentence will be replaced with 'surface layers'.

L248: Please introduce figure 13 first and explain the DO abbreviations before going into the analysis.

Change: will do.

Fig. 6: Can the authors speculate why their biogeochemical model is unable to quantitatively reproduce the observed oxygen minimum? What processes do they think the model misses?

Response: We believe that the insufficient oxygen depletion as suggested by Fig. 14 (probably this the one the referee is referring to, and not Fig. 6) might be associated with the inaccuracies in benthic consumption rates. A model that considers the horizontal heterogeneities in the soil permeability, and that dynamically calculates the vertical profiles in the benthic layer could potentially better reproduce the oxygen consumption rates. In order to prevent any potential misunderstanding (see our response to the 'Review overview' above) we would like to clarify once again, that the model does have a benthic component based on the benthic model of ECOHAM. This is however a simple model that dynamically tracks the nutrient and carbon pools only, and the benthic DO consumption rate is computed based on a linear relationship with benthic remineralization, based on empirical evidence (see Paetsch and Kühn 2008).

Change: we will include a discussion along this line in the text.

Fig. 15: These again are too small and I cannot see the arrows at all in the difference figures.

Change: we will reorganize the figure.

L282: Yes, they do but this is rather an open door. Any reader would have expected that from the start, and would have been surprised if this was not the case.

Response: that 'the efficiency of estuarine circulation is determined by an interplay between the meteorological and hydrological conditions' may be an intuitive expectation, but we are not aware of any previous study that provided evidence to support this intuition. Nevertheless, the word 'indicate' potentially implies 'novelty', which was not intentional.

Change: we will reformulate and expand the sentence.

Sec4: Please discuss the lack of bacterial dynamics in the discussion, and the effect this can have on the simulated results.

Response: we would like to clarify that, although the presented model does not account for the bacteria biomass, the primary function of bacteria in the context of the current study, at least as represented in biogeochemical models (such as ECOHAM), i.e., decomposition of DOM and the resulting DO consumption (probably this is what concerns the referee, based

on their comment under 'Review overview') is represented in our model by a first order kinetic term (Fig. 2, Table B9). Conceptually, this is equivalent to assuming that the degradation of DOM is not limited by bacterial biomass. We are not aware of any evidence against this assumption for the study area. For the case of Lake Kinneret, Li et al. (2014) have shown that the DO estimates of a model version similar to ours, that also does not explicitly describe bacterial biomass 'were not significantly different' than those estimated by two other model variants where bacterial dynamics were explicitly described. In conclusion, we do not see the need for an extensive discussion of the lack of an explicit description of bacterial dynamics.

L312: I would say the model was able to reproduce the physical characteristic features of the system quite well.
Response: we believe we provide evidence for the ability of model to reproduce several non-physical characteristic features of the system.

L316: "The skill of the model ... is notable", quite a nice notation as it is meaningless. Notable means it can be noted, it says nothing about it being good or bad.
Change: we will expand this in relation to Helgoland being at a transition zone, and that the reproduction of certain signals, such as the summer peak in DIN being dependent on reproduction of the spread of the freshwater plume.

L320-333: I'm not sure why this is include here, this is not of interest for the general reader I would think. Therefore I would put this in an appendix at most.
Response: we believe that this paragraph is necessary, as it provides a perspective in relation to the recent modeling studies, and points to the important trade-off between computational expense and performance, which should be relevant virtually for anyone who is interested in coupled physical-biogeochemical modeling.

L349: I fail to see the prolonged stratification in figure 11. As these are all July averages I don't see a time indication in this figure at all.
Change: the sentence will be reformulated as 'uninterrupted phases of stratification during July, that gave rise to a large average density difference (Fig. 11), … '.

L385: I object to the use of the word "satisfactorily" when it comes to the reproduction of the biogeochemical features of the German Bight ecosystem.
Response: we are convinced that the coupled model system satisfactorily reproduces a number characteristic features of the ecosystem, that are relevant for the purposes of this study.

Table B5,B6,B8: If parameter values are provided then references on what these are based on should be included as well. Assuming these values have not been published before.
Change: we will provide the sources of parameters, where possible and necessary.

**Language**
In general I found the manuscript very readable, yet the English used was not always correct or as expected. I found several mistakes regarding single/plural (e.g. L 140, "the discharge rates ... peaks", L156 "Comparison ... are shown", L299 "potential sources of error needs to be addressed"), omissions of articles (e.g. L 156 A "Comparison of", L187 "Despite a tendency to overshoot, the range of", L205 The "Effect of exchanging", L211 "further to the North"), additions of articles in unnecessary places (e.g. L 143 "over the central Europe", LL232 "river forcing of the 2012 is used", L232 "the plume of the DIP")

and omission of connecting words (e.g. L157 "are located at shallow sites, and therefore provide", L373 "the presence of regional differences"). I suggest the authors check their English thoroughly before the next submission. But I love the double negative found on L439: "leading to near-complete elimination of negative values of the total mixing being removed". So the removal has been eliminated?

Change: we will fix the mistakes pointed out and will check the manuscript once again and try to eliminate further potential mistakes.

**References**

Fennel, K., Laurent, A. (2018) N and P as ultimate and proximate limiting nutrients in the northern Gulf of Mexico: implications for hypoxia reduction strategies. Biogeosciences 15, 3121–3131.

Große, F. (2017) The influence of nitrogen inputs on the oxygen dynamics of the North Sea. PhD Thesis, University of Hamburg. Pages 82-83.

Große, F., Kreus, M., Lenhart, H.-J., Pätsch, J., and Pohlmann, T. (2017) A Novel Modeling Approach to Quantify the Influence of Nitrogen Inputs on the Oxygen Dynamics of the North Sea, Frontiers in Marine Science, 4, 1–21.

Irby, ID., Friedrichs, M.A.M., Da, F., Hinson, K.E. (2018) The competing impacts of climate change and nutrient reductions on dissolved oxygen in Chesapeake Bay. Biogeosciences 15, 2649–2668.

Kerimoglu, O., Hofmeister, R., Maerz, J., Riethmüller, R., and Wirtz, K. W. (2017a) The acclimative biogeochemical model of the southern North Sea, Biogeosciences 14, 4499–4531.

Kerimoglu, O., Jacquet, S., Vinçon-Leite, B., Lemaire, B. J., Rimet, F., Soulignac, F., Trévisan, D., and Anneville, O. (2017b) Modelling the plankton groups of the deep, peri-alpine Lake Bourget, Ecological Modelling, 359, 415–433.

Li, Y., Gal, G., Makler-Pick, V., Waite, A. M., Bruce, L. C., and Hipsey, M.R. (2014) Examination of the role of the microbial loop in regulating lake nutrient stoichiometry and phytoplankton dynamics. Biogeosciences, 11, 2939–2960.

Nasermoaddeli, M. H., Lemmen, C., Stigge, G., Kerimoglu, O., Burchard, H., Klingbeil, K., Hofmeister, R., Kreus, M., Wirtz, K. W., and Kösters, F. (2018) A model study on the large-scale effect of macrofauna on the suspended sediment concentration in a shallow shelf sea, Estuarine, Coastal and Shelf Science 211, 62-76.

Pätsch, J., Kühn, W. (2008) Nitrogen and carbon cycling in the North Sea and exchange with the North Atlantic—A model study. Part I. Nitrogen budget and fluxes. Continental Shelf Research 28, 767-787.

Pätsch, J., Burchard, H., Dieterich, C., Gräwe, U., Gröger, M., Mathis, M., Kapitza, H., Bersch, M., Moll, A., Pohlmann, T., Su, J., Ho-Hagemann, H. T., Schulz, A., Elizalde, A., and Eden, C. (2017) An evaluation of the North Sea circulation in global and regional models relevant for ecosystem simulations, Ocean Modelling, 116, 70–95.

---

## Author Comment (AC2) · 29 Apr 2020

**Response to Anonymous Referee #2**

Below, referee comments (starting with 'Comment'), and our specific responses (starting with 'Response' and/or [envisioned] 'Change') are provided in black and blue fonts, respectively.

Comment: This study introduces a new biogeochemical model, consisting of modified versions of previously published models. Coupled with a hydrodynamic model with an improved mixing scheme, the system is validated in the German Bight region, and used to assess the impact of meteorology and river forcing on a specific flood event in 2013. The conclusion is that an interplay of the two resulted in anomalous conditions, as previously noted in observations.
The paper acts as both presentation and validation of a new modelling system, and an investigation into a specific event. While it could potentially work as separate papers, the paper is well enough written and laid out, and the validation both sufficiently comprehensive and targeted, that it works well and is an enjoyable and interesting read. I recommend publication in Biogeosciences subject to a few minor comments detailed below.
Response: We thank the referee for the careful read of the manuscript, positive assessment of our work and constructive suggestions.

The biogeochemical model appears to be a work in progress towards a different mixotroph-based model, rather than a model likely to be widely used in its present form, if I've got the correct impression? This is fine, given its structure seems sensible and plenty of validation is presented, but it would be worth adding some discussion about what sets it apart from other similar models, particularly ECOHAM, and what future developments are intended.
Response: The Referee's impression is correct, that the model presented in this study is intended to be developed further. Nevertheless, we also believe that at its present state, it can already serve the purposes of this study.  We agree that further clarification and additional discussion of the model structure and future directions is needed.
Change: we will clarify the similarities and differences of our model with similar models, particularly ECOHAM in the revised manuscript, and discuss the potential directions for further model development.

Given that it's central to the study, in Section 2.1 and/or 2.2 it would be worth explicitly detailing which variables are used in the atmospheric and riverine forcing, and how they're applied to the model (e.g. bulk formulae? are rivers applied just at the surface or over the full depth?).
Change: further details on the application of meteorological and riverine forcing in the model will be provided.

"using the 'spatial.cKDTree' package from the Scipy library of Python 3.5." – add the Scipy version number for completeness.
Change: the Scipy version that will be used for the revised manuscript will be provided.

Figure 3 shows the Ems, and that this doesn't have anomalous discharge in 2013. This isn't mentioned or discussed in Section 3.1, and should be. Also, the "dashed blue lines" appear solid.
Change: we will mention  which rivers were affected and which were not  in the discussion, and correct the caption of Fig. 3.

"Simulated temperature and salinities . . . (Fig. 5) . . . exhibit no signs of systematic deviations or biases." The calculated B* values are near-zero, but by eye it looks like there's a cold bias, particularly at colder temperatures, and that salinity is usually too high. Is this just a trick of the eye, or are the simulated and observed distributions different? Please also state what B*, rho and n are in the caption of Fig. 5, as per Fig. 6.

Response: A careful assessment of the figure reveals that a slight cold bias at the lower range is indeed present, which seems to be canceled out by the slight warm bias at the higher range. But these deviations are mostly within a 1K range, therefore presumably do not have a significant effect. At an intermediate range, salinity is somewhat (in the order of 2 g/kg) overestimated, indicating insufficient spread of coastal waters with low salinity. This may either be due to (still) underestimated horizontal mixing, or inaccuracies in the advection patterns. Either way, the potentially underestimated salinity during the studied event may lead to an underestimation of the importance of riverine discharges on the stratification dynamics in the transition zone characterized by intermediate (29-32 g/kg) salinities.

Change: a more nuanced description of the model performance, and implications thereof will be provided in the discussion. Definition of B*, rho and n will be included in caption of Fig.5.

"(Fig.8) . . . The ability of the model to capture the sharp increase in DIN during June/July 2013 at the Helgoland station suggests that the spreading of the plume of the Elbe-Weser rivers following the flood event was realistically reproduced." The model completely misses the peak earlier in the year, and also in early 2014. Can you be confident therefore that this result was obtained for the right reasons?

Response: The sentence was indeed misleading, as the word 'suggest' emphasizes the uncertainties of mechanisms causing the summer peak. The reasons for not reproducing the peaks in DIN during winters are not clear, but the reason for the mid-July peak as captured by the model is very likely to be the flood. Occurrence of such a high summer DIN peak at this station is not common under typical hydrological settings. In this particular case, we can tell with certainty that the reason for the model to produce such a high summer peak is the flood: the Figure R2-1 below shows how the flood water characterized by low-salinity and high DIN move within the 45 days after the flood event. These findings are consistent with the *in-situ* data shown by Voynova et al. (2017, Fig. 12), building confidence to believe that the unique DIN peak measured at Helgoland in July 2013 was caused by the flood in reality as well.

Change: the figure shown below might be too specific for the manuscript, but this finding will be more clearly described in the text.

[Figure]

*Figure R2-1: Salinity and DIN concentrations following the flood event. Arrow shows the location of monitoring station at Helgolands.*

In Fig. 7, plotting the average in white is confusing – I initially thought there were separate yellow/blue lines either side of it, and the white was blank space. Plotting it in dark yellow/blue might be better. Also, make clear in the caption that the line indicates the average and the shading the standard deviation (I assume this is the case?).
Change: dark yellow/blue lines will be used to show the averages, and it will be clarified that the dark lines indicate averages and shades indicate standard deviations (indeed).

In Fig. 14 it would be best to avoid plotting red and green together, as this renders it inaccessible to those who are red-green colour blind. (Disclaimer: I'm not colour blind myself, so can't say for sure.)
Change: a colorblind-friendly palette will be used in Fig.14.

**References**

Voynova, Y. G., Brix, H., Petersen, W., Weigelt-Krenz, S., and Scharfe, M. (2017) Extreme Flood Impact on Estuarine and Coastal Biogeochemistry: the 2013 Elbe Flood, Biogeosciences, 14, 541–557.

---

## Referee Report (RR1)

**Reviewer 1 response to author's comments**

**GENERAL**

I am a little bit confused in general by the authors response, which seems directed purely at me and which includes references to changes in the future tense ("we will"). My confusion on certain descriptions can be shared by others, and therefore the authors should not explain things directly to me but in the manuscript text. And they should state clearly where they have included new or altered text, for instance "in line xxx we have now included the following text: …" so that reviewers like me can see what alteration have been included. Statements like "we will extend the model description section to clarify the model structure" without stating the actual changes made do not allow me to judge the changes made to the manuscript to assess improved readability. Now I had to search for changes in the resubmitted text, which did not include track changes of any form, in order to locate any alterations. These seemed to be semantical throughout, with the only real addition in the new section 4.3 Model limitations and perspectives, where the authors have included some discussion on some of the reservations I had towards the applied model.

Model limitations are better included now, in said section, including the lack of bacterial dynamics and sediment nutrient budgets. Figures have been enlarged and are much more readable, and there is a better model description in the manuscript, outlining differences with existing models. However, the impact of benthos is still not discussed and the authors remain, in my view, rather optimistic about the performance of the biological part of their model. They did downscale their conclusions on the biological impact somewhat by stressing that the results are indicative of a system response, rather than a predicted ecosystem state.

Some detailed responses are given below.

1. "It is not entirely clear to us, what is meant by 'why not use ECOHAM for that'. If it means directly applying a readily available ECOHAM setup, this was not an option: to the best of our knowledge, the only available HAMSOM-ECOHAM setup, performance of which have been sufficiently documented (e.g., Große, 2017), is simply too coarse (20 km horizontal resolution, and 7 z-levels within the deepest part of the study area) for being able to capture the meso-scale features of the system, in particular, the haline stratification caused by the Elbe river (Pätsch et al., 2017)."

I was not referring to an existing setup (an application of a model to a certain area), but to a model itself (ECOHAM), therefore I do not understand the given response: horizontal and vertical resolutions can be adapted to create a more suitable setup. It may not have been practical (is there a FABM version of ECOHAM?), but that is another reason. It seems to me the choice of model was based on availability and previous experience, rather than a careful process of selecting the best suitable model for the desired application. This is fine, and common practice in a world limited by project deadlines and budgets, but it should be stated as such. Including the limitations of this model compared to others. Within the new manuscript the model is now compared to others applied in the same area, which is somewhat limited.

**5. "It was again not entirely clear to us what is exactly meant by 'a model with benthos' by the referee."**

I am a bit mystified by the authors answer. They state that they do not understand the line " a model with benthos", and then proceed to to detail some of the abiotic benthic processes included in the model. Even if by rare chance they really do not know the meaning of the word "benthos" (organisms living in, on or near the seabed) it is easy to look it up. And they should explain in the manuscript the importance of benthic organisms in sediment dynamics and seabed nutrient budgets, and the possible impact of not including these dynamics on their results. In shallow coastal areas the benthic compartment forms an integral part of the local ecosystem, including its carbon and nutrient pathways. At the moment this is not included in the added section 4.3. The impact of burrowing or filter feeding animals on water quality is not negligible. Neither are algal mats preventing resuspension.

Response: There are certainly more complex models, but considering the purposes of our study, it is not clear in which specific sense would such a model be better suited. It should be noted that, with regard to benthic/pelagic coupling, models of similar complexity have been used until recently, for studying the nutrient concentrations and bottom oxygen conditions in the North Sea (e.g., Große et al., 2017, using ECOHAM), as well as other similarly dynamic coastal shelf systems such as the Louisiana Shelf (e.g., Fennel and Laurent, 2018) or even shallower systems such as the Chesapeake Bay (Irby et al. 2018).

**Response: necessity for the presentation of a validation of benthic nutrient concentrations of fluxes (which is very rarely done in studies similar to ours) is not clear.**

There are, as said, more complex models out there. The question is whether they do a better job of describing the pelagic-benthic coupling and resulting pelagic nutrient concentrations in shallow areas, by allowing for bio-turbidity, bio-irrigation, filter feeding and long-tern storage of nutrients and as such creating more dynamic sediment nutrient profiles and benthic-pelagic nutrient fluxes. Validation is indeed seldom done, and it is hard to get "right" compared to in-situ measurements (which are mere snapshots), but that doesn't mean it shouldn't be done. It would be good to know if the fluxes as produced by the model are comparable to observations, to get an idea of 1. The importance of benthic-pelagic fluxes in the area for pelagic nutrient concentrations (particularly in stratified conditions) and 2. The (possible) limitations of the simple benthic module.

**L201 "A consistent source of error seems to be the failure of the model to estimate the timing of the spring bloom"**

This is a serious issue and I appreciate the authors frankness regarding it. It is difficult to get the timing correct, but in essence the main aim of a coastal biogeochemical model is just that: to assess/predict/be able to analyse coastal productivity. To have this as a consistent error is worrying and suggests the biogeochemical model is missing some key drivers / processes, as indicated by the lack of good Chl validation results. Which is why I do not share the confidence of the authors in the presented biogeochemical model.

**L417 "oxygen consumption and denitrification (Fig. 2) were largely based on ECOHAM (see Section 2.1 Appendix B), which was earlier derived from ERSEM."**

The ERSEM model is mentioned a few times in the text, but only with the references of Ford et al (2017) and Blackford and Radford (1995). In line 417 it is mentioned for the first time without any references at all. I strongly suggest including the original ERSEM publications by Baretta and Baretta-Bekker, in order to give credit where credit is due:

Baretta JW, Ebenhöh W, Ruardij P (1995) The European Regional Seas Ecosystem Model, a complex marine ecosystem model. Neth J Sea Res 33(3/4):233–246

Baretta-Bekker JG (ed) (1995) European Regional Seas Ecosystem Model I (1990–1993). Neth J Sea Res 33(3/4):229–483

Baretta-Bekker JG, Baretta JW (eds) (1997) European Regional Seas Ecosystem Model II (1993–1996). J Sea Res 38(3/4):169–436

**L445 "which is out of the scope of this study."**

I have never heard the combination "out of the scope of" before, "outside of the scope of" is more common.

**L457 "However, the model structure and formulations represent the state of the art"**

I find this very bold, given the fact that the benthic compartment is mainly parameterized, mixotrophs are not included, bacterial dynamics are not directly included and different algal groups are not represented (e.g. the mentioned *Phaeocystis*). The model is not bad, and will be near state of the art with the additions and developments mentioned in the text. But this is not the model applied here.

---

## Author Response (AR2)

**Response Overview and List of Changes**

Below, comments of the editor and referee (starting with '**Comment:**'), and our responses (starting with 'Response' and/or 'Change') are provided in black and blue fonts, respectively.

Dear Dr. Cook,

we would like to thank the referee for their further comments and you for your re-evaluation.

Regarding the points that you highlighted in your report;

**Comment:** 1. When explaining or clarifying a point made by the reviewer, it should not just be aimed at the reviewer, it should also include any necessary change to the text, and a clear indication of where this has occurred. As the reviewer notes this was not always clear in your response.

Thank you for pointing this out. We felt that some of the referee's comments required clarification, but we were under the impression that these would not be relevant for the readership. We would also like to clarify, that in the previous revision round, we had included marked-up version with our revisions, and where we saw the need, we had indicated the sections where the changes had taken place. Based on these materials we submitted, we assumed that it would be straightforward to locate where the changes had taken place. In our point-to-point responses below, we indicated the line numbers of some of the relevant changes that we had introduced in the previous revision. In order to prevent such inconveniences for this revision, we included the line numbers that indicate where the changes took place in the newly revised manuscript.

**Comment:** 2. That there be some comparison with the benthic fluxes put out by the model with actual measured rates, ideally within the same system with reference to the role of benthic organisms may have over this.

We have addressed this issue, by further elaborating the benthic fluxes and including an extensive discussion of the role of benthic organisms.

**Comment:** 3. Tone down some of the overconfident language about the accuracy of the model.

As we documented in detail below, in our previous revision, we already had changed and where necessary, expanded a number of statements that we considered being potentially overconfident. In the current revision, we removed the only specific statement pointed out in the respective comment of the referee in the detailed comments section.

Our point-to-point responses, and a marked-up manuscript version can be found below.

On behalf of all authors,
Onur Kerimoglu

**Response to Anonymous Referee #1**

GENERAL
**Comment:** I am a little bit confused in general by the authors response, which seems directed purely at me and which includes references to changes in the future tense ("we will"). My confusion on certain descriptions can be shared by others, and therefore the authors should not explain things directly to me but in the manuscript text. And they should state clearly wherethey have included new or altered text, for instance "in line xxx we have now included the following text: ..." so that reviewers like me can see what alteration have been included. Statements like "we will extend the model description section to clarify the model structure" without stating the actual changes made do not allow me to judge the changes made to the manuscript to assess improved readability. Now I had to search for changes in the resubmitted text, which did not include track changes of any form, in order to locate any alterations.

Response: We would like to clarify that, in our response letter (which is based on, but not identical to our response in discussion forum), where we used past tense, we did refer to the sections where the change occurred, whenever we felt that the location of the change may not be obvious. We also did submit a marked-up manuscript version, according to the instructions specified in the journal's web site. We verify that, at the time of writing this response, we can access both of these documents through the respective website of the journal. For some comments, we were under the impression that an explanation to the referee seemed more suitable than a change in the manuscript, since we were not convinced that these would relevant for the general audience. In our current response, we will refer to line numbers to indicate any changes in the manuscript, in hopes to prevent any inconvenience.

**Comment:** These seemed to be semantical throughout, with the only real addition in the new section 4.3 Model limitations and perspectives, where the authors have included some discussion on some of the reservations I had towards the applied model.

Response: There were various other changes, which have been documented in our earlier response letter and the marked-up manuscript version.

**Comment:** Model limitations are better included now, in said section, including the lack of bacterial dynamics and sediment nutrient budgets. Figures have been enlarged and are much more readable, and there is a better model description in the manuscript, outlining differences with existing models. However, the impact of benthos is still not discussed

Response: In our previous revision, we had added a full paragraph in section 4.3 on 'the role of sediments', as was requested by the referee. See our response below about the discussion of benthos (a discussion of which was previously not explicitly requested).

**Comment:** and the authors remain, in my view, rather optimistic about the performance of the biological part of their model. They did downscale their conclusions on the biological impact somewhat by stressing that the results are indicative of a system response, rather than a predicted ecosystem state.

Response: in the previous revision, besides the change referred to by the referee, we had included a dedicated section (4.3) for discussing the limitations of the approach, and had toned down any statements that may be regarded as overconfident. In particular:
1) We had replaced 'and exhibit no signs of systematic deviations or biases' with 'There is a slight cold bias at the lower temperature range (29-32 g/kg), which seems to be canceled out by the slight warm bias at the higher range. These deviations are mostly

within a 1 °C range, therefore presumably do not have a significant effect. At an intermediate range, salinity is overestimated by up to 2 g/kg, indicating insufficient spread of coastal waters with low salinity. This may either be due to (still) underestimated horizontal mixing (see Appendix A), or inaccuracies in the advection patterns.' (L168-172).

2) We had included the explanation 'Underestimated NO3 at an intermediate range (10-40 µMN) is possibly due to the aforementioned underrepresentation of N-rich riverine waters within the transition zone. Regarding DIP, the measured-simulated pairs that represent major underestimation errors (e.g., in the <0.5 µMP simulation band) point to the inability of the model to capture summer maxima occurring in specific coastal regions.' (L174-177).

3) Relevant to points 1 and 2, in the discussion, we had included: 'The insufficient spread of coastal waters is potentially the reason for overestimated salinities and underestimated NO3 in the transition zone, characterized by intermediate salinities and NO3 concentrations (Fig. 5). These errors, in turn, may have lead to an over- and underestimation of the importance of riverine discharges on the stratification dynamics and productivity in the coastal and transition zones, respectively (L316-319).

4) In the original statement 'the oxygen depletion in the bottom layers in 2013, and the lack thereof in 2012 are *qualitatively reproduced*.', we had replaced '*qualitatively reproduced*' with 'qualitatively captured, although the DO depletion in 2013 is not fully reproduced' (L283).

5) We had replaced '… low bias and high correlation coefficients in general (e.g., Fig. 5)' with '… low bias and high correlation coefficients for temperature, salinity and nutrients (e.g., Figs. 5, 6, 8). The skill of the model in reproducing chlorophyll concentrations was not as good (Fig. 8, see below for a discussion of potential reasons)' (L328-330).

6) We had removed the oversimplifying statement ' The ability of the model to capture the sharp increase in DIN during June/July 2013 at the Helgoland station suggests that the spreading of the plume of the Elbe-Weser rivers following the flood event was realistically reproduced' in the Results section (3.2) and provided the following, nuanced discussion: 'For instance, the sharp DIN peak observed and simulated at Helgoland during June/July 2013 is uncommon for the summer season (see Fig.12 in Voynova et al., 2017). Overlapping DIN and freshwater fronts simulated by the model, temporarily spreading to the west of Helgoland during the same period (not shown), and supported also by a sharp decline of observed and simulated salinities (Fig. 6), reveal that this rare summer DIN peak was caused by the plume of the Elbe-Weser flood. This provides evidence for the model's ability to reproduce the behavior of the plume.' (L334-339).

Please see below for the additional actions taken in this revision, in response to a specific comment of the referee.

**Comment:** Some detailed responses are given below.
1. "It is not entirely clear to us, what is meant by 'why not use ECOHAM for that'. If it means directly applying a readily available ECOHAM setup, this was not an option: to the best of our knowledge, the only available HAMSOM-ECOHAM setup, performance of which have been sufficiently documented (e.g., Große, 2017), is simply too coarse (20 km horizontal resolution, and 7 z-levels within the deepest part of the study area) for being able to capture the meso-scale features of the system, in particular, the haline stratification caused by the Elbe river (Pätsch et al., 2017)."
I was not referring to an existing setup (an application of a model to a certain area), but to a model itself (ECOHAM), therefore I do not understand the given response: horizontal

and vertical resolutions can be adapted to create a more suitable setup. It may not have been practical (is there a FABM version of ECOHAM?), but that is another reason.

Response: In our previous response, besides clarifying why using an existing ECOHAM setup was not an option, we had also provided a detailed explanation of the shortcomings of ECOHAM regarding the description of phytoplankton physiology and growth, and how these are more realistically represented in our model by accounting for their variable Chl:C:N:P composition. To the best of our knowledge, and based on our personal communication with H. Lenhart, J. Pätsch, M. Kreus, there is no functional FABM version of ECOHAM (an initiative several years ago was discontinued).

**Comment:** It seems to me the choice of model was based on availability and previous experience, rather than a careful process of selecting the best suitable model for the desired application. This is fine, and common practice in a world limited by project deadlines and budgets, but it should be stated as such. Including the limitations of this model compared to others. Within the new manuscript the model is now compared to others applied in the same area, which is somewhat limited.

Response: The referee's impression that the choice of model was based on the availability of our experience and other external motivations is wrong. There are good reasons to keep the model complexity low, such as avoiding overparameterization and reducing computational costs.

We compared our model against four recent studies that describe model applications in the study region and four other model studies in different, but comparable systems. We believe that these references provide a sufficient perspective, and show that in some aspects, even simpler models, in comparison to ours, are being actively developed and used for studying the research questions similar to those we addressed in this study.

**Comment:** 5. "It was again not entirely clear to us what is exactly meant by 'a model with benthos' by the referee." I am a bit mystified by the authors answer. They state that they do not understand the line " a model with benthos", and then proceed to to detail some of the abiotic benthic processes included in the model. Even if by rare chance they really do not know the meaning of the word "benthos" (organisms living in, on or near the seabed) it is easy to look it up. And they should explain in the manuscript the importance of benthic organisms in sediment dynamics and seabed nutrient budgets, and the possible impact of not including these dynamics on their results. In shallow coastal areas the benthic compartment forms an integral part of the local ecosystem, including its carbon and nutrient pathways. At the moment this is not included in the added section 4.3. The impact of burrowing or filter feeding animals on water quality is not negligible. Neither are algal mats preventing resuspension.

Response: these comments are connected with the next set of comments, therefore please see our response below.

**Comment:** Response: There are certainly more complex models, but considering the purposes of our study, it is not clear in which specific sense would such a model be better suited. It should be noted that, with regard to benthic/pelagic coupling, models of similar complexity have been used until recently, for studying the nutrient concentrations and bottom oxygen conditions in the North Sea (e.g., Große et al., 2017, using ECOHAM), as well as other similarly dynamic coastal shelf systems such as the Louisiana Shelf (e.g., Fennel and Laurent, 2018) or even shallower systems such as the Chesapeake Bay (Irby et al. 2018).

Response: necessity for the presentation of a validation of benthic nutrient concentrations

of fluxes (which is very rarely done in studies similar to ours) is not clear.

There are, as said, more complex models out there. The question is whether they do a better job of describing the pelagic-benthic coupling and resulting pelagic nutrient concentrations in shallow areas, by allowing for bio-turbidity, bio-irrigation, filter feeding and long-tern storage of nutrients and as such creating more dynamic sediment nutrient profiles and benthic-pelagic nutrient fluxes. Validation is indeed seldom done, and it is hard to get "right" compared to in-situ measurements (which are mere snapshots), but that doesn't mean it shouldn't be done. It would be good to know if the fluxes as produced by the model are comparable to observations, to get an idea of 1. The importance of benthic-pelagic fluxes in the area for pelagic nutrient concentrations (particularly in stratified conditions) and 2. The (possible) limitations of the simple benthic module.

Response:

1) We believe that 'getting an idea of the importance of benthic-pelagic fluxes in the area for pelagic nutrient concentrations' requires a targeted study, and is beyond the scope of our study.

2) In the previous revision of our manuscript, we had already mentioned limitations cast by the simple benthic model, such as the underestimated phosphorus concentrations (L463-469) and underestimated oxygen depletion (L425-L429), as driven by potentially misrepresented POM deposition rates (L431-434) and benthic oxygen profiles (L460-463).

Change: In the newly revised version, we substantially extended the discussion of the benthic processes by: a) mentioning the estimated relevance of benthic transformations in the study system (L422-425) b) pointing to the link between the benthic oxygen consumption and remineralization (L429-431) rates; c) a detailed account of the role of benthos on benthic fluxes, and implications of not having resolved these in our model (L434-449); d) a comparison of estimated and measured benthic denitrification rates L(450-459).

**Comment:** L201 "A consistent source of error seems to be the failure of the model to estimate the timing of the spring bloom"

This is a serious issue and I appreciate the authors frankness regarding it. It is difficult to get the timing correct, but in essence the main aim of a coastal biogeochemical model is just that: to assess/predict/be able to analyse coastal productivity. To have this as a consistent error is worrying and suggests the biogeochemical model is missing some key drivers / processes, as indicated by the lack of good Chl validation results. Which is why I do not share the confidence of the authors in the presented biogeochemical model.

Response: We believe that the failure of our model to estimate the timing of the spring bloom does not disqualify it from assessing *the response of coastal productivity to the hydrometeorological extremes during the summer months, and the mechanisms mediating this response,* which we targeted in this study. Secondly, as we had pointed out in section 4.3, the most likely reason for this failure is understood and its implications were discussed in the previous revision of the manuscript (L368 and L473-474).

**Comment:** L417 "oxygen consumption and denitrification (Fig. 2) were largely based on ECOHAM (see Section 2.1 Appendix B), which was earlier derived from ERSEM."

The ERSEM model is mentioned a few times in the text, but only with the references of Ford et al (2017) and Blackford and Radford (1995). In line 417 it is mentioned for the first time without any references at all. I strongly suggest including the original ERSEM publications by Baretta and Baretta-Bekker, in order to give credit where credit is due:

Baretta JW, Ebenhöh W, Ruardij P (1995) The European Regional Seas Ecosystem

Model, a complex marine ecosystem model. Neth J Sea Res 33(3/4):233–246

Baretta-Bekker JG (ed) (1995) European Regional Seas Ecosystem Model I (1990–1993). Neth J Sea Res 33(3/4):229–483

Baretta-Bekker JG, Baretta JW (eds) (1997) European Regional Seas Ecosystem Model II (1993–1996). J Sea Res 38(3/4):169–436

Change: at the first mentioning of ERSEM, we now cited the first reference suggested by the referee (L418). But we believe that citing two further suggested journal volumes is not warranted, considering that these are not cited even by the recent ERSEM studies, and as their relevance for the scientific argumentation of the manuscript are not clear.

**Comment:** L445 "which is out of the scope of this study."
I have never heard the combination "out of the scope of" before, "outside of the scope of" is more common.
Change: we adopted the suggested form (L477).

**Comment:** L457 "However, the model structure and formulations represent the state of the art" I find this very bold, given the fact that the benthic compartment is mainly parameterized, mixotrophs are not included, bacterial dynamics are not directly included and different algal groups are not represented (e.g. the mentioned Phaeocystis). The model is not bad, and will be near state of the art with the additions and developments mentioned in the text. But this is not the model applied here.
Change: We removed the statement and in the following sentence, we replaced 'this study is of heuristic value' with '
[revised manuscript text omitted]